

# ClimaMeter: Contextualising Extreme Weather in a Changing Climate

Davide Faranda[1,2,3], Gabriele Messori[4,5,6], Erika Coppola[7], Tommaso Alberti[8], Mathieu Vrac[1], Flavio Pons[1], Pascal Yiou[1], Marion Saint Lu[1], Andreia N. S. Hisi[1,11], Patrick Brockmann[1], Stavros Dafis[9,10], and Robert Vautard[1]

[1]Laboratoire des Sciences du Climat et de l'Environnement, UMR 8212 CEA-CNRS-UVSQ, Université Paris-Saclay, IPSL, 91191 Gif-sur-Yvette, France
[2]London Mathematical Laboratory, 8 Margravine Gardens London, W6 8RH, UK
[3]Laboratoire de Météorologie Dynamique/IPSL, École Normale Supérieure, PSL Research University, Sorbonne Université, École Polytechnique, IP Paris, CNRS, Paris, 75005, France
[4]Department of Earth Sciences, Uppsala University, Uppsala, Sweden
[5]Swedish Centre for Impacts of Climate Extremes, Uppsala University, Uppsala, Sweden
[6]Department of Meteorology, Stockholm University, Stockholm, Sweden.
[7]The Abdus Salam International Center for Theoretical Physics, Trieste, Italy.
[8]Istituto Nazionale di Geofisica e Vulcanologia, Rome, Italy.
[9]National Observatory of Athens, Institute for Environmental Research and Sustainable Development, I. Metaxa &Vas. Pavlou, P. Penteli (Lofos Koufou), 15236 Athens, Greece.
[10]Data4Risk, Paris, France.
[11]Sorbonne Université, Paris, France.

**Correspondence:** Davide Faranda (davide.faranda@cea.fr)

**Abstract.** Climate change is a global challenge with multiple far-reaching consequences, including the intensification and increased frequency of many extreme weather events. In response to this pressing issue, we present ClimaMeter, a platform designed to assess and contextualise extreme weather events relative to climate change. The platform offers near real-time insights into the dynamics of extreme events, serving as a resource for researchers, policymakers, and being a science dissemination tool
for the general public. ClimaMeter currently analyses heatwaves, cold spells, heavy precipitation and windstorms. This paper elucidates the methodology, data sources, and analytical techniques on which ClimaMeter relies, providing a comprehensive overview of its scientific foundation. To illustrate Climameter, we provide four examples, the December 2022 North American Winter Storm, the August 2023 Guangdong - Hong Kong Flood, the late 2023 French Heatwave and the July 2023 windstorm Poly. They underscore the role of ClimaMeter in fostering a deeper understanding of the complex interactions between climate
change and extreme weather, with the hope of ultimately contributing to informed decision-making and climate resilience.

## 1 Introduction

The consequences of climate change are becoming increasingly evident and widespread, making the need for a comprehensive and timely understanding of their current and future implications acute (Allan et al., 2021; Hartin et al., 2023). A number of recent high-impact extreme weather events, such as the 2021 North American heatwave (Philip et al., 2021; Lucarini et al.,



2023), the 2019-2020 wildfires in Australia and the 2023 wildfires in Canada (Bowman and Sharples, 2023), the 2021 Ahr floods in Germany and Benelux (Cornwall, 2021), the summer and autumn drought of 2020 in the Southwest USA (Dannenberg et al., 2022) and the 2020 North Atlantic Hurrícane season (Reed et al., 2022), have once again raised to the forefront the potential role of climate change in making extreme weather more frequent and severe. As climate records are repeatedly shattered, a crucial task for scientific understanding, climate policy-making and communication to the general public is to dis-

tinguish between those extreme events primarily issuing from natural variability and those which were significantly modulated by climate change-related factors (Trenberth, 2011; Trenberth et al., 2015; Mahony and Cannon, 2018; Huggel et al., 2016). As a result, a number of rapid tools for linking extreme events to climate change have emerged (e.g., Stott et al., 2016; Angélil et al., 2014; Otto, 2016; Otto et al., 2018; Vautard et al., 2018; Faranda et al., 2022) . These tools bridge the gap between the realms of climate science, policy and public awareness, offering a means to decipher the complex web of interactions between

human-induced climate change and extreme weather events.

Here, we present a new step in the direction of effective and real-time assessment of extreme weather events in the context of climate change: the ClimaMeter platform, available at www.climameter.org. ClimaMeter provides a near real-time assessment of extreme weather, balancing the competing needs of rapidity, accuracy and accessibility for a broad public. Specifically, the platform offers a non-technical report with a summary figure, intended for the general public and media, and more detailed

supplementary figures providing additional analysis, aimed at fellow researchers. A defining feature of ClimaMeter is its accessibility, allowing users to explore and visualize results through an intuitive and user-friendly interface. ClimaMeter currently analyses heatwaves, cold spells, heavy precipitation and windstorms.

ClimaMeter has been developed as a collaborative effort among climate scientists, meteorologists, and data analysts, and harnesses state-of-the art historical weather reconstructions and statistical algorithms based on dynamical system metrics to

determine the influence of climate change on specific extreme weather events. This ClimaMeter core group is responsible for selecting and analyzing extreme events, producing reports, and addressing media inquiries. In its initial phase of expansion, ClimaMeter welcomes scientists interested in investigating the relationship between extreme events and climate change, inviting them to join this important endeavor. The aim is to fulfill the need for a near-real time understanding of the interactions between climate change, natural variability, and specific extreme weather events.

Classical extreme event attribution techniques usually depend on a single variable, and do not consider the extreme event as a dynamically evolving weather system, which may influence several meteorological variables, with potentially compounding effects. For instance, cyclones and storms can lead to impacts from strong winds, pluvial floods and storm surges (e.g., Alberti et al., 2023). A framework like ClimaMeter, which views the extreme weather events as synoptic-scale weather systems, has so far been missing.

In the remainder of this study, we provide a detailed explanation of the methodological foundations of ClimaMeter, present report writing protocols for extreme events and user-oriented features of the ClimaMeter website. Finally, we show four examples of ClimaMeter extreme weather reports: the Guangdong - Hong Kong floods on 2023/09/08, the late French Heatwave on 2023/08/20-23, Storm Poly affecting Northern Europe on 2023/07/05, and the US winter storms of 2022/12/21-27.



## 2 Methodology

Our methodology is based on looking for weather conditions similar to those that caused the extreme event of interest (atmospheric circulation analogues, Yiou, 2014; Faranda et al., 2020) with statistical algorithms based on dynamical system metrics. The object studied (i.e., "the event") is a surface-pressure pattern over a certain region and averaged over a certain number of days, that has led to extreme weather conditions. The analysis of an event is decided upon within the ClimaMeter core-group based on national and international media reports of societal, economic and/or environmental losses, or if the event in ques-

tion had unique features from a meteorological perspective. The geographical area and time-period for analysing the event are determined based on the locations of the above-mentioned impacts and on a visual analysis of the meteorological drivers and surface footprint of the event. For example, in the case of a summer heatwave associated with an atmospheric block, we would select a region including both the block and the land areas affected by the highest temperatures. The final choice is based on expert judgment, following an open discussion in the ClimaMeter core-group.

We focus on the satellite era, namely the period since 1979, when continuous observations of climate variables from satellites have become available (e.g. Hersbach et al., 2018). We consider the early decades of the satellite era (1979–2000, "past") and the more recent decades (2001–2022, "present") separately. The "past" is meant to be representative of a world with a weaker anthropogenic influence on climate than the "present", which refers to present-day conditions strongly affected by anthropogenic climate change. We use data from MSWX (Beck et al., 2022), freely available in real-time at https://www.gloh

2o.org/mswx/, but in this article we also show results obtained analyzing ERA5 data (Hersbach et al., 2015) (see Appendix A). We then compare weather conditions associated with analogues in the two periods, and test for significant changes. In other words, given the definition of the event, any change in the probability or intensity of meteorological hazards (e.g., extreme rain or heat) will be conditioned to the atmospheric circulation (e.g., Vautard et al., 2016; Shepherd, 2016; Yiou et al., 2017). Next, we evaluate whether the observed changes in the extreme event, if any, are likely due to natural variability or anthropogenic

climate change.

Since we use publicly available historical climate reconstructions instead of numerical model simulations, the framework is rapid, reproducible and independent of model bias. However, our approach also comes with disadvantages. In some cases, the extreme event can result from very unusual weather situations that may not have previously occurred in the analysis period. In this case, the identified atmospheric circulation analogues will be poor, and the confidence we place in our results is low.

Moreover, the use of 1979–2000 as reference "past" period comes with the risk of underestimating the role of anthropogenic climate change, as this period cannot be viewed as a stationary, unforced climate.

### 2.1   Data Download and Pre-processing

1. The latest available data from the MSWX-Past data product are downloaded. If necessary, these are completed with the MSWX-NRT (MSWX near real-time) product (Beck et al., 2022). Specifically, we download surface pressure, 2-

80       m temperature, total precipitation and 10-m wind speed data at daily resolution. As a gridded meteorological product, MSWX-NRT does not reflect extremes at spatial scales smaller than the grid's size. Moreover, the analysis data used





for the near real-time extension might suffer from model errors. For these reasons, extreme values of temperature, precipitation and wind speed measured locally by meteorological stations and mentioned in our reports, may not be reflected in the MSWX-NRT data shown in our analysis.

2. The event is represented by a surface pressure pattern averaged over a certain number of days ($\geq 1$) and over a certain geographical region. These are determined through a consensus expert judgement (see Sect. 2 above). We use surface pressure as MSWX does not currently provide mid-tropospheric fields, and other reanalysis products which provide these, do so with a considerable time delay.

3. Surface pressure and 2-m temperature data are pre-processed by removing, at each grid point and for each day, the average of their values for all the corresponding calendar days over the period 1979–2022. This accounts for the seasonal cycle and, only for surface pressure data, this also removes the effect of varying surface elevation in space. Total precipitation and wind speed data are not pre-processed. When more than one day is considered, a moving average across the event duration is performed.

4. Similar past events are searched by looking for *analogues* in terms of the event's surface pressure pattern only, over the selected spatio-temporal domain (see Sect. 2 above). We divide the surface pressure data set into the previously-mentioned "past" and "present" periods, and look for analogues in each period separately. Once analogues are found, we compute the corresponding temperature, precipitation or wind speed composite maps based on the best 15 analogues in each period. That is, the surface pressure fields minimizing the Euclidean distance to that of the event itself. For the present period, the event itself is excluded from the present composite maps. In addition, we exclude analogues within a window of 21 days centered on the date of the event (central date for events that last for longer than one day). When varying the number of selected analogues between 10 and 20 for the case-studies presented later in this study, we do not find any qualitatively important differences in our results. Ginesta et al. (2023) performed extensive robustness tests of the methodology with respect to changes in the number of selected analogues, changes in the geographical domain's extension and changes in the temporal duration analysed for an event.

5. We next compute differences between the composite surface pressure and temperature, precipitation or wind speed fields for analogues in the two periods. We additionally compute differences in composite temperature, precipitation and wind speed for analogues in the two periods in three selected major urban areas within the analysis region.

6. In order to evaluate the possible role of low-frequency modes of natural variability in explaining the differences between the composite maps of analogues in the two periods, we also include in our analysis monthly indices from NOAA/ERSSTv5 for the El Niño-Southern Oscillation (ENSO), the Atlantic Multidecadal Oscillation (AMO), and the Pacific Decadal Oscillation (PDO). We compare the distributions of the ENSO, AMO and PDO values on the dates of the analogues in the past and present periods, and we test the statistical significance of the observed differences. If a significant different is found, we consider that the mode of variability could possibly influence the observed changes





in the analogues of the event between the two periods. This step permits an uncomplicated first assessment of the possible influence of natural variability, yet comes with several limitations. First, we only use three amongst the many large-scale climate variability modes known from the literature. Moreover, not finding a significant difference between the distributions of an index conditionally to an event's analogues in the two periods does not guarantee the absence of an effect from that variability mode. For example, a resurgent or transitioning La Nina is linked to significant shifts in patterns of convective outbreaks over the U.S.A. (Lee et al., 2016), yet these ENSO phases are not caught by the ENSO3.4 index used in our analysis. Finally, in our analysis we always give equal weight to all three modes, even though depending on the geographical location of the event being analysed some of the modes may be more relevant than others. The ENSO and AMO data are freely retrieved from the Royal Netherlands Meteorological Institute (KNMI) Climate Explorer (`https://climexp.knmi.nl/start.cgi`), and the PDO time series from the National Centers for Environmental Information (NCEI) of the National Ocean and Atmospheric Administration NOAA (`https://www.ncei.noaa.gov/`), where the most updated version is available.

## 2.2 Visual Representation

Figure 1 is a schematic explanation of the summary ClimaMeter figure, present in all extreme weather reports.

The top row of the figure consists of two gauge charts (Figure 1, upper panels). The left-hand-side one indicates the respective roles of natural variability and climate change in explaining the changes detected in the event (i.e., "strengthened (for cold spells: weakened) by climate change" or "influenced by natural variability"). The right-hand-side gauge indicated the rarity of the surface-pressure pattern of the event (i.e., "the event is unique" or "similar events have occurred in the past"). The gauge representation is a visually immediate way to communicate this complex information. The gauge needles can take four positions: almost entirely to the left (5%), 2/3 to the left (35%), 2/3 to the right (65%), or almost entirely to the right (95%). These categories are determined based on the values of underlying quantitative metrics.

Furthermore, we also provide visual representations of the surface pressure anomalies of the event and of the hazard variables, i.e. temperature, precipitation or wind speed (Figure 1, top middle panels). We also provide corresponding maps of the composite differences between analogues in the "past" and "present" periods (bottom middle panels). Finally, we report the seasonality of the analogues in each period (Figure 1, lower left panel), and detected changes in temperature, precipitation and wind speed in three major urban areas within the analysis domain (lower right panel). More in detail:

1. To determine the influence of natural variability or climate change on the event (left-hand gauge), we look at whether the analogues in the two periods occurred during significantly different phases of the ENSO, and/or the AMO, and/or the PDO. If none of the three modes shows significant differences, then the dial points 95% to the right. For each statistically significant difference in one of the variability modes, we shift the dial 30% to the left. Since we consider three modes, the dial can thus attain values of 95%, 65%, 35% or 5%. We do not use 0% or 100% to acknowledge data and analysis uncertainties.





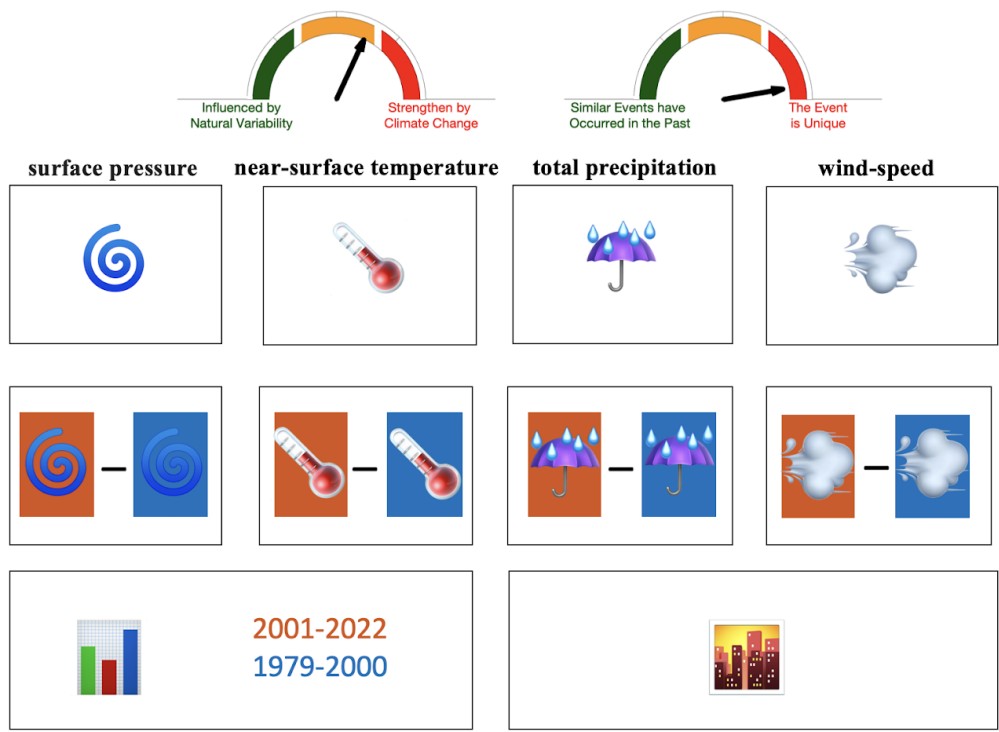

**Figure 1.** Schematic illustration of the ClimaMeter figure output. The top row of the figure consists of two gauge charts. The left-hand-side one indicates the respective roles of natural variability and climate change in explaining the changes detected. The right-hand-side gauge indicated the rarity of the surface-pressure pattern of the event. The second row provides a visual representations of the surface pressure anomalies of the event and those of the hazard variables, i.e., temperature, precipitation and wind-speed. The third row provides maps of the composite differences between analogues in the "past" and "present" periods. The last row reports the seasonality of the analogues in each period (left panel), and detected changes in temperature, precipitation and wind speed in three major urban areas within the analysis domain (right panel). See Section 2.2 for more details.

2. Concerning the rarity of the event in the data record (right-hand gauge) we use analogue quality Q, previously introduced in Faranda et al. (2022). In our analysis, $Q$ is simply the average Euclidean distance of a given day from its 15 closest analogues.

   (a) If for both the past and the present, $Q$ is below the 75th percentile of the distribution of $Q$ values computed for all days in each period, the gauge points left (5%). This means that similar events have occurred in the past.

   (b) If, instead, for both the past and the present periods, $Q$ is between the 75th and the 95th percentiles, we assign 35%.

   (c) If for the past or the present $Q$ is between the 75th and the 95th percentiles, while for the other period it is above the 95th percentile, we assign 65%.

150



(d) If $Q$ exceeds the 95th percentile for both the past and the present, we assign the maximum value to the gauge (95%). This means that the event is unique in our dataset.

We choose relatively high percentiles of $Q$ to determine the positioning of the dial since we analyse extreme events which, by their very nature, are not frequently observed. The dial should therefore be interpreted as referring to events which in any case are comparatively unusual, but which may not be unique. As for the other dial, we do not use 0% or 100% to acknowledge data and analysis uncertainties.

3. We display the event's average surface pressure anomaly, defined as the difference between the average surface pressure at each gridbox in the selected region for the duration of the event and the average surface pressure at each gridbox for the same calendar day(s) over the whole period 1979–2022. The same is displayed for hazards (i.e., temperature, precipitation or wind speed).

4. We also display the difference between the average surface pressure for all analogues in the present period and the average surface pressure for all analogues in the past period. The same is done for the selected hazard variable. To determine significant changes between the two periods, we adopt a bootstrap procedure which consists of pooling the dates from the two periods together, randomly sampling 15 dates from this pool 100 times (higher values do not change significantly the results), creating the corresponding difference maps and marking as significant only grid point changes larger than two standard deviations above or below the mean of the bootstrap sample. This is implemented in the summary figure for the surface pressure changes but not for the hazard changes. For the latter, significance is only included in the extended analyses (see Appendix A) and is highlighted in the report text.

5. To assess the significance of the differences in the indices of natural variability during analogues in the past and present periods, we use a two-tailed Cramér-von Mises test at the 0.05 significance level. If the p-value is smaller than 0.05, the null hypothesis that both samples are from the same distribution is rejected, namely we interpret the distributions as being significantly different.

## 3 Report Writing Protocol

ClimaMeter has a structured protocol for writing reports that assess and contextualise extreme weather events relative to climate change. This is a living document, that can be updated based on input from the ClimaMeter core team. The latest version of the template at the time of writing, which is detailed in Appendix A1, encompasses all aspects of the report, including the formulation of the report title, homepage title (which appears on the ClimaMeter.org homepage when the report is released), press summary, event description, climate and data background, ClimaMeter analysis, and conclusion.

The homepage title categorizes extreme weather events into those "strengthened by human-driven climate change", "mostly strengthened by human-driven climate change", "likely influenced by both human-driven climate change and natural variability," or "mostly driven by natural variability". This dichotomy is chosen to provide a clear and immediate communication, even



though we appreciate that there may be factors affecting the extremes which fall in neither category (for example human-driven land-use changes). In the report itself, the template starts with a press summary that provides context for the event, assesses its uniqueness, and characterizes it in terms of the role of climate change versus natural variability, based on the ClimaMeter analysis. The event description section details the specifics of the extreme weather event, including dates, location, and key characteristics. Links to relevant media reports are provided. It also presents a simplified explanation of the atmospheric

conditions leading to the event. The climate and data background section refers to IPCC reports and other relevant scientific information to provide context and support for the analysis. It also assesses the confidence level in the analysis based on the uniqueness or not of the event (right-hand dial). A unique event is associated with low confidence, since analogues will be poor; an event similar to others observed in the past gives us a higher confidence in the analogues-based analysis. The ClimaMeter analysis section examines changes in surface pressure and hazard variables, comparing the present and past analogues to

determine how the event has evolved between the two periods. It further evaluates the extent to which climate change may have strengthened the event (left-hand dial). The conclusion provides a two-sentence summary of the analysis, summarising the two dials and the analogue composite difference maps. Overall, this protocol aims to offer an accessible yet comprehensive approach to assessing extreme weather events in the context of climate change, prioritising a clear communication of findings.

The report for a given extreme event is released as soon as possible after the event has occurred, if needed using the MSWX-

200 NRT data (see Sect. 2.1). When updated MSWX-Past data become available, we aim to update the report. Similarly, we update reports based on feedback from researchers and the general public. This means that we may update a given report several times. For every report, we indicate the first publication date, the date of the latest update, and the date when the report was finalised. Once the report is finalised, it cannot be changed further. We will provide a PDF version of the report as well as a DOI to cite the report. The finalisation of the report may happen several months after the occurrence of the extreme event being analysed.

## 205 4 ClimaMeter Website

The website www.climameter.org comprises several pages, each serving distinct functions as listed below.

– The **Home** provides real-time updates and summaries of the most recent extreme weather event reports, with links to the full reports.

– The **Event Dashboard** provides an at-a-glance overview of all the extreme events that have been analyzed. Users can

navigate the extreme events visually, based on the location and type of event – visualised with colour-coded icons on a world map. The dashboard also enables users to refine their search by filtering events according to location, time range, event category, or the sources of detected changes. A screenshot of the dashboard is provided in Figure 2.

– The **Hazard Database** page provides the full list of extreme events that have been analyzed, divided by category. At the time of writing, the categories are "Heatwaves", "Cold Spells", "Heavy Precipitation" and "Windstorms".





– The **Methodology** page offers an overview of the scientific methods employed by ClimaMeter. It explains the underlying principles, data sources, and analytical techniques. This serves as a reference for researchers and climate scientists seeking insights into the scientific basis of ClimaMeter's conclusions.

   – The **About ClimaMeter** page outlines the project's origins, goals, and the core team responsible for ClimaMeter's operation.

– The **Media Coverage** page compiles news articles, interviews, and reports featuring ClimaMeter. It tracks the impact of ClimaMeter on public fora and climate-related decision-making processes.

   – The **Peer-Reviewer Research** provides a list of investigations from research efforts within peer-reviewed journals.

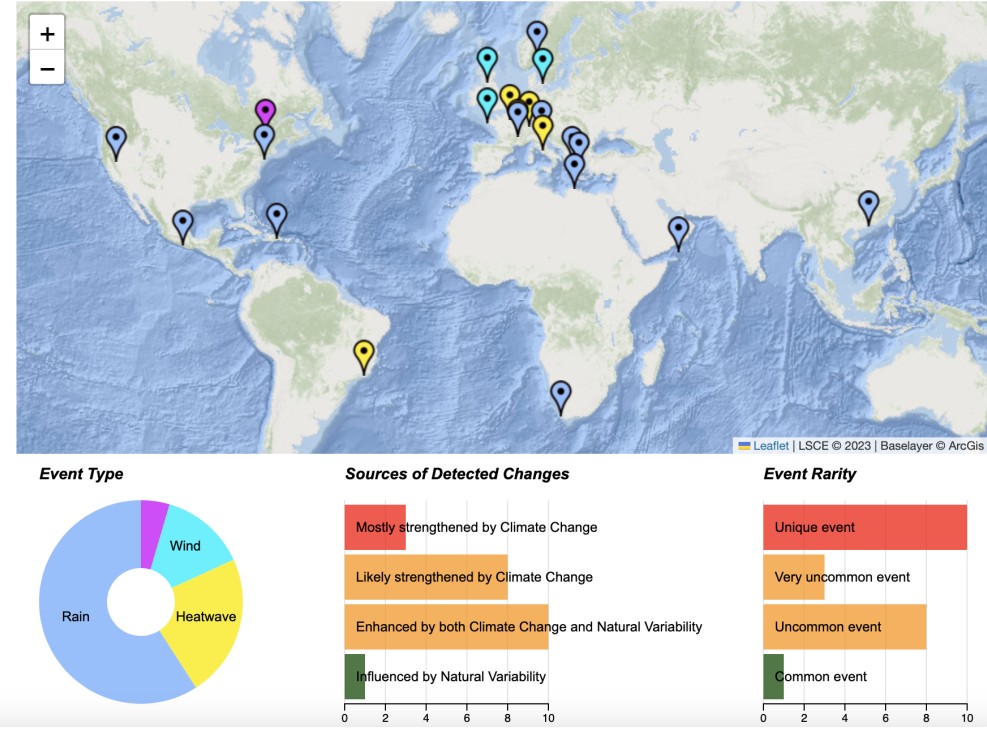

**Figure 2.** Screenshot of the Event Dashboard appearing in the ClimaMeter website, with extremes analyzed up to end of November 2023. See https://www.climameter.org/event-dashboard.

## 5 ClimaMeter Report Examples

In the following, we present four examples, illustrative of the extreme event categories that ClimaMeter reports on. These are
selected to offer a broad geographical coverage. The results are adapted from existing ClimaMeter reports in order to fit the





format of a scientific paper. They are presented following the ClimaMeter protocol structure, but may not follow the protocol line-by-line due to the above-mentioned adaptations.

## 5.1 2022/12/21-26 North American Winter Storm [-110°E -45°E 30°N 66°N]

**Event Description.** In late December 2022, a low pressure area wreaked havoc across the U.S.A. and parts of Canada.
From December 21st to 26th, the storm resulted in blizzards, high winds, heavy snowfall, and record-breaking cold temperatures wikipediaDecember2022. The affected regions included Minnesota, Iowa, Wisconsin, Michigan, Ohio, Pennsylvania, New York, and Ontario, with some areas enduring nearly two days of zero-visibility blizzard conditions. The storm also impacted other parts of the northeastern U.S., resulting in significant snowfall and high tides. Ocean-effect snow hit Cape Cod, and in Canada, cities like Kingston, Prince Edward County, and Fort Erie experienced blizzard conditions and record-breaking wind gusts. The cold wave reached as far south as Miami, Florida, causing wind chill alerts for 110 million people across 36 U.S.A. states. The storm and the accompanying cold wave claimed the lives of at least 100 people. The Buffalo area in New York was particularly hard-hit, with lake-effect snowfall exceeding 140 cm over five days, leading to 41 fatalities. The storm also caused multiple road accidents, road closures, and flight cancellations. Buffalo Niagara International Airport was shut down for five days, and rail services were severely disrupted. Power outages affected approximately 6.3 million households in the U.S.A. and 1.1 million in Canada. The storm was unofficially named Winter Storm Elliott by The Weather Channel and was described by the National Weather Service as a "once-in-a-generation storm" for Buffalo (National Weather Service Buffalo, NY, 2023). NOAA's Weather Prediction Center deemed it a "historic arctic outbreak," and it was widely referred to as the "Blizzard of the Century" (CNN, 2022). The storm's meteorological history began on December 21, with the low pressure strengthening over the Northern Plains and intensifying into a bomb cyclone. As its central pressure plummeted, the storm caused intense winds and drew bitterly cold air along. The Surface Pressure Anomalies (Figure 3) reveal a weather pattern characterized by a low-pressure systems over Labrador Peninsula. This unusual atmospheric setup led to a significant influx of cold air directed toward the Great Lake region and Eastern US. Temperature Anomalies indicate that most of the area covered by the analysis experienced extremely cold anomalies reaching up to -20°C in certain areas. Precipitation Data show that the Eastern Coast of Canada and USA and Quebec received large amounts of precipitation, mostly in the form of snow. The storm also caused strong winds as shown in Windspeed Data.

**Climate and Data Background for the Analysis.** The IPCC AR6 WG1 report (Seneviratne et al., 2021) discusses the impact of climate change on the frequency and intensity of cold outbreaks in North America. The continent's warming trend is not uniform across, but exhibits pronounced polar amplification, particularly during the winter months. Extreme high-temperature records are now being set more frequently than extreme cold records due to anthropogenic climate change, and there is a generalised reduction in the severity of extreme cold events. The IPCC further highlights that the effects of climate change extend to other climatic components linked to low temperatures, including reductions in snow cover, glacier extent, and sea and lake ice.

Our analysis approach rests on looking for weather situations similar to those of the event of interest having been observed in the past. For the December 2022 North American Winter Storm, we have low confidence in the robustness of our approach





given the available climate data, as the event is largely unique in the data record.

**ClimaMeter Analysis.** Figure 3 reports ClimaMeter results on the December 2022 North American Winter Storm and how events similar to it have changed in the present (2001–2022) compared to what they would have looked like if they had occurred in the past (1979–2000). The Surface Pressure Changes show that low pressure systems have not significantly changed their

intensity compared to the past, with significant signals restricted to regional scales. Temperature Changes show that similar events produce milder temperatures in the present than in the past, with anomalies ranging over 0 – +5 °C. The urban areas of Montreal, New York and Detroit see an increase in temperature in the present period, although the changes are not statistically significant for Montreal. Montreal and New York also receive more precipitation in the present period than in the past. We do not find any large shift in seasonality of the events, with "Similar Past Events" being more frequent in December and February

in the present period, while in the past they were mostly occurring in January and March. Finally, we find that sources of natural climate variability, notably the El Niño-Southern Oscillation and the Atlantic Multidecadal Oscillation, may have influenced the event. This suggests that the changes we see in the event compared to the past may be partly due to human driven climate change, with a contribution from natural variability.

**Conclusions.** Based on the above, we conclude that low pressure patterns similar to that causing the December 2022 North American Winter Storm have become 0–5 °C warmer in the present than in the past. We interpret the December 2022 North American Winter Storm as a largely unique event for which natural climate variability played a role.

### 5.2    2023/09/08 Guangdong - Hong Kong Floods [108°E 120°E 18°N 25°N]

**Event Description.** On September 8th, 2023, the Guangdong and Hong Kong regions in China experienced severe flooding due

to torrential rains (FloodList, 2023). This extreme weather, associated with the remnants of Typhoon Haikui, led to widespread flooding and significant damage (Guardian, 2023). The Hong Kong Observatory indicated that the Pearl River Estuary was affected by a low pressure, resulting from cyclone Haiku, and causing persistent heavy rain and thunderstorms. Rainfall exceeded 600 mm at the Observatory, and the Eastern and Southern Districts of Hong Kong Island saw rainfall totals higher than 800 mm, far exceeding the monthly average for September, which is typically 321.4 mm. The Observatory Headquarters recorded

a record-breaking one-hour rainfall of 158.1 mm, the highest ever recorded since records began in 1884. This deluge caused streets to transform into rivers, with floodwaters over 1.5 meters deep. The consequences were severe, with disrupted road traffic, public transport, and subway stations. Schools, the Hong Kong Stock Exchange, and some offices were forced to close, and only essential travel was advised by the Labor Department. The flooding also caused extensive damage to residential and commercial buildings, including a popular shopping mall. In Hong Kong, at least two fatalities were reported as a result of the

flooding, over 100 people were injured and hundreds of people were evacuated. The Home Affairs Department swiftly opened 14 temporary shelters to accommodate those displaced by the disaster. The impact of the flooding extended beyond Hong Kong, affecting the nearby Guangdong province, particularly areas in Shenzhen and Meizhou, where approximately 11,000 people had to be evacuated. Shenzhen also recorded an unprecedented 469 mm of rain in a 24-hour period. Surface Pressure



30–Nov–2023CNRS–IPSL

## ClimaMeter for North American Winter Storm
## 21-Dec-2022 to 27-Dec-2022

**Figure 3.** ClimaMeter output for 2022/12/21-26 North American Winter Storm. See Figure 1 for an explanation of the different panels.





Anomalies show a weak negative pressure anomaly over the western part of the domain analysed, associated with Precipitation

of over 100 mm/day (Figure 4).

**Climate and Data Background for the Analysis.** According to the Intergovernmental Panel on Climate Change (IPCC) WG2 Chapter 10 (Shaw et al., 2022), Hong Kong has already been experiencing noticeable changes in its precipitation patterns, particularly those exceeding 100 mm of rainfall in a single day (R100). These changes are associated with a projected decrease

in the annual number of rain days, alongside with a shift in the seasonality of rainfall, with more prolonged wet seasons. However, it is crucial to acknowledge that future projections exhibit substantial variability among different model and emission scenarios, underscoring the significant uncertainties that persist.

Our analysis approach rests on looking for weather situations similar to those of the event of interest having been observed in the past. For the Guangdong and Hong Kong floods we have low confidence in the robustness of our approach given the

available climate data, as the event is largely unique in the data record.

**ClimaMeter Analysis.** Figure 4 reports ClimaMeter results on the Guangdong - Hong Kong floods and how events similar to them have changed in the present (2001-–2022) compared to what they would have looked like if they had occurred in the past (1979-–2000). The Surface Pressure Changes show that cyclones leading to the Guangdong - Hong Kong floods are

slightly more intense (1 hPa) in the present period than they were in the past. Precipitation Changes show that similar events produce larger (up to 5 mm/day) amounts of precipitation in the Western part of the region in the present period. However, they also produce weaker precipitation in other areas, such as over Hong Kong itself. Indeed, the Hong Kong urban area experienced a decrease in precipitation in the present period of almost 3 mm/day. The urban area of Dongguan, in Guangdong province, experienced a corresponding increase, while Shenzhen shows little change. We also find that "Similar Past Events"

have become more frequent in September, and less common in October, possibly indicating a shift in the seasonality of this type of events. Finally, we find that sources of natural climate variability, notably the Atlantic Multidecadal Oscillation, may have only partly influenced the event. This suggests that the changes we see in the event compared to the past may be mostly due to human driven climate change.

**Conclusion.** Based on the above, we conclude that pressure patterns leading to the Guangdong - Hong Kong floods, similar to that observed in September 2023, are up to 5 mm/day wetter in the western Guangdong province than they would have been in the past and up to 3 mm/day dryer in the coastal part of the analysis domain. We interpret the Guangdong - Hong Kong floods as an event whose characteristics can mostly be ascribed to human driven climate change.

### 5.3  2023/08/20-23 Late Summer French Heatwave [-10°E 20°E 30°N 52°N]

**Event Description.** Starting from August 20th, Western and Northern Europe experienced unusually high temperatures, peaking on the 22nd and 23rd August. Extremely high temperatures were recorded at many locations: with a national temperature indicator of 27.5°C, Wednesday August 23rd was the second-hottest day ever recorded in French history (TF1 Info, 2023). A





30–Nov–2023CNRS–IPSL

## ClimaMeter for Guangdong - Hong Kong Floods
## 08-Sep-2023

**Figure 4.** ClimaMeter output for 2023/09/08 Guangdong - Hong Kong Floods. See Figure 1 for an explanation of the different panels.





large number of daily maximum temperature records were broken. In Toulouse, for example, the thermometer reached 42.4 °C (previous record was 40.7 °C). Others were greatly surpassed, like in Auch, reaching 42.3 °C compared to the previous 40.9

°C, and in Narbonne, where the mercury climbed to 42.1 °C from the previous record of 39.8 °C. Additionally, the heatwave was extreme also in mountain areas with Aiguille du Midi ( 3800 m.a.s.l. in the Mont-Blanc massif) recording over 10 °C maximum temperature. Finally, the minimum daily temperature of 30.4 °C in Menton, set a new absolute national record for minimum daily temperature in mainland France. This heatwave ended on August 24th, when fresher air from the Atlantic reaching the country caused severe thunderstorms.

The heatwave was associated with a persistent area of high pressure (anticyclone) over Western and Central Europe, on the background of a warm Atlantic ocean and warm Mediterranean sea and of a positive phase of the El Niño–Southern Oscillation. The Surface Pressure Anomalies (Figure 5) pattern associated with the event consists of a high pressure area over the Alps and Central Europe and a low pressure area over the Atlantic. Temperature Anomalies show that temperatures were 10°C or more warmer than usual over most of the domain considered (Figure 5).


**Climate and Data Background for the Analysis.** Chapter 11 of the IPCC AR6 report (Seneviratne et al., 2021) emphasizes that in Western Europe, there is strong evidence indicating a very likely increase in maximum temperatures and the frequency of heatwaves. Specifically, in Western Europe, climate warming has already reached 1.7°C compared to the pre-industrial era over the last decade, with 1.5°C of this increase occurring since the 1960s, particularly during the summer months. The number

of heatwave days in Western Europe has multiplied by five, transitioning from an annual average of 2 days between 1960 and 2020 to about 10 days.

For this event, we have low confidence in the robustness of our approach given the available climate data, as the event is largely unique in the data record.

**ClimaMeter Analysis.** Figure 5 reports ClimaMeter results on the late Summer French Heatwave and how events similar to this have changed in the present (2001--2022) compared to what they would have looked like if they had occurred in the past (1979--2000) in the region [-10°E 20°E 30°N 52°N]. Surface Pressure Changes show that the pressure over the mountainous areas of the Alps, and Massif Central has become higher, favouring a more intense warming due to solar radiation and vertical motions of the air. Temperature Changes show that similar events produce temperatures which in the present climate are

between 1 ºC and 4 ºC hotter than what they would have been in the past. This coincided with temperatures in Lyon, Toulouse and Marseille being over 1.5 ºC hotter than what they would have been in the past. We also note that Similar Past Events have become more common in the month of August, while they previously occurred largely in July. However, the differences in average maximum temperatures between these two months are limited in many French cities.

Finally, we find that sources of natural climate variability, notably the Atlantic Multidecadal Oscillation, may have partly

influenced the event. This means that the changes we see in the event compared to the past may be due to human driven climate change.



**Conclusions.** Based on the above, we conclude that heatwaves similar to the late August 2023 French heatwave have become between 1 °C and 4 °C warmer in the present than in the past. We interpret this heatwave as a largely unique event whose characteristics can be mostly ascribed to human driven climate change.

**Figure 5.** ClimaMeter output for 2023/08/20-23 Late Summer French Heatwave. See Figure 1 for an explanation of the different panels.





### 5.4 2023/07/05 Storm Poly in Northern Europe [0°E 25°E 46°N 60°N]

**Event Description.** On July 5th 2023, a summer extratropical storm named Poly hit Germany, the Netherlands and Denmark, causing significant damage and resulting in 2 casualties. It featured hurricane-force wind gusts locally up to 146 km/h, the strongest ever recorded for a summer storm in the Netherlands (EUMETSAT). The storm's rapid cyclogenesis began over the North Atlantic. Once it made landfall, severe winds were accompanied by heavy rainfall and led uprooted trees and transportation disruptions. The majority of severe weather reports associated with the storm concerned intense wind events. Storm Poly displayed clear negative Surface Pressure Anomalies over the Netherlands, Denmark and parts of North-Western Germany, while wind speed during the storm in the MSWX data we use for analysis, was above 30-40 km/h over a large swath of the Baltic and Northern Europe (Figure 6).

**Climate and Data Background for the Analysis.** In Chapter 11 of the IPCC AR6 report (Seneviratne et al., 2021), it is highlighted that there is low confidence in recent total extratropical storm changes globally, but medium confidence in a poleward storm track shift since the 1980s. Understanding past-century extratropical storm trends is hindered by interannual variability and variations in the assimilated data, particularly when moving from the pre-satellite era to the satellite era. Data for the Northern Hemisphere supports a decreased central pressure for cyclones ($< 970$ hPa) in summer and winter during 1979-–2010, but with non-monotonic trends. The background mean sea level pressure's seasonal and regional variations complicate assessing extratropical storm intensity trends based on absolute central pressure. Our analysis approach rests on looking for weather situations similar to those of the event of interest having been observed in the past. For Poly, we have low confidence in the robustness of our approach given the available climate data, as the event is largely unique in the data record.

**ClimaMeter Analysis.** Figure 6 reports ClimaMeter results on the storm Poly and how events similar to this have changed in the present (2001-–2022) compared to what they would have looked like if they had occurred in the past (1979-–2000). The Surface Pressure Changes show that the pressure over the storm area has become lower, favouring more intense cyclones in the present period than in the past. wind speed Changes show that similar events produce winds which are between 2 and 6 km/h stronger than what they would have been in the past, consistent with the Surface Pressure Changes. Storms similar to Poly are associated with stronger winds in Hamburg (Germany) and Copenhagen (Denmark) than they would have been in the past. We also note that Similar Past Events have become more common in the month of August, while they previously occurred more in other summer months (peaking in July) or even in September. Finally, we find that sources of natural climate variability, notably the Pacific Decadal Oscillation, may have only partly influenced the event. This means that the changes we see in the event compared to the past may be mostly due to human driven climate change.

**Conclusion.** Based on the above, we conclude that storms similar to Poly display lower pressures and stronger winds in the present than in the past. We interpret this storm as a largely unique event whose characteristics can mostly be ascribed to human driven climate change.



30–Nov–2023CNRS–IPSL

# ClimaMeter for Storm Poly
## 05-Jul-2023

**Figure 6.** ClimaMeter output for 2023/07/05 Storm Poly in Northern Europe. See Figure 1 for an explanation of the different panels.





## 6   Conclusions

ClimaMeter is a pioneering effort to contextualise the ever-increasing occurrence of extreme and hazardous weather events across the globe relative to the ongoing human-driven climate change. This responds to a pressing need to enhance the way we communicate on this critical issues to the general public and to provide a new tool for policymakers who face the implications of climate change.

At the heart of ClimaMeter's approach lies an analysis of weather conditions similar to those that caused the extreme event of interest (so-called atmospheric circulation analogues), diagnosed using surface pressure. The analysis then leverages analogues to diagnose changes over time in four key meteorological hazard indicators: rainfall, wind speed, and high and low temperatures. These, combined with the analysis of the circulation analogues, serve as the cornerstone for understanding and contextualizing the dynamics of the analysed extreme weather phenomena.

ClimaMeter's framework is not restricted to a specific region or event, as evidenced by the four case examples presented in Sect. 5. These underscore the significance of contextualising extreme events, as a tool to understand the broader context within which they occur and the interplay between changes in their dynamics and changes in the associated hazards. This is a crucial step in understanding how climate change is influencing specific extremes.

As the world grapples with an escalating series of climate-related challenges, it is our hope that ClimaMeter may emerge as a useful tool for a variety of stakeholders. Researchers can utilize ClimaMeter to gain deeper insights into the relationship between climate change and specific extreme events, even for events which may not be the object of a conventional scientific article. Policymakers can rely on ClimaMeter for evidence as to how and to what extent specific extreme event categories in a given geographical area are affected by climate change, thus providing a knowledge basis for addressing the growing risks and vulnerabilities associated with extreme events. Finally, the general public can benefit from ClimaMeter's accessible and informative approach, fostering greater awareness and understanding of the urgency and complexity of climate change and its consequences. This occurs both directly through ClimaMeter's website and social media and indirectly, through the media reports on ClimaMeter analyses.

Ultimately, we hope that frameworks like ClimaMeter may be one small piece in the puzzle to achieve a more resilient and sustainable climatic future, by integrating scientific research, communication and operational implementation.

## Appendix A: ClimaMeter template for reports

### A1   Template for report titles

For the report titles we use the following formulations. We have removed here the case of cold spells to make the text easier to follow.

- «Heavy precipitation/high temperatures/strong winds» in «location name» strengthened by human-driven climate change [if both gauge indicators are on the red]





- «Heavy precipitation/high temperatures/strong winds» in «location name» mostly strengthened by human-driven climate change [if one of the gauge indicator is in red and the other is yellow]

- «Heavy precipitation/high temperatures/strong winds» in «location name» likely influenced by both human-driven climate change and natural variability [if both gauge indicators are in the yellow or one is in the green and one in the yellow]

- «Heavy precipitation/high temperatures/strong winds» in «location name» mostly driven by natural variability [if both gauge indicators are on the green]

- Low confidence prevents ascribing «Heavy precipitation/high temperatures/strong winds» in «location name» to human-driven climate change [if the left gauge indicator is green and the right gauge indicator is red]

## A2    Press Summary

For the first sentence about changes in the event intensity:

- «event type» similar to «event name» are «change here» in the present than they would have been in the past «geographical area here» [example: Cold spells similar to Borea are 2 ℃ warmer in the present than they would have been in the past across all of Northern Europe].

For the second sentence about the uniqueness of the event:

- «event name» was a largely unique event [if arrow of right-hand-side-dial points to the right]

- «event name» was a very uncommon event [if arrow of right-hand-side-dial points 3/4 to the right]

- «event name» was a somewhat uncommon event [if arrow of right-hand-side-dial points 3/4 to the left]

- «event name» was a similar to several events in the past [if arrow right-hand-side-dial points to the left]

For the third sentence about the role of climate change vs natural variability:

- We ascribe the «high/low/heavy/strengthened» «variable name here» of/associated with «event name» to human driven climate change and natural climate variability likely played a minor role [if arrow of left-hand-side-dial points to the right].

- We mostly ascribe the «high/low/heavy/strengthened» «variable name here» of «event name» to human driven climate change and natural climate variability likely played a modest role [if arrow of left-hand-side-dial points 3/4 to the right].

- Natural climate variability likely played a role in driving the pressure pattern and the associated «increase/decrease» in «variable name here» linked to «event name», but human-driven climate change may have also contributed [if arrow of left-hand-side-dial points 3/4 to the left].





    – Natural climate variability likely played an important role in driving the pressure pattern and the associated «increase/decrease» in «variable name here» linked to «event name» [if arrow of left-hand-side-dial points to the left].

## A3    Event Description

«On/Starting from/In the period» «date(s)», «location» experienced «brief description of event, ideally with some numbers» [example: unusually low temperatures for the season, with $-20°C$ being recorded in Stockholm. These frigid temperatures were part of a broader area of below-average temperatures, peaking in the first week of December and stretching from Scandinavia all the way to Southern France]. «The «event type»/ During «event name» or similar» «brief description of the atmospheric configuration for laymen» [example: During the Scandinavian cold spell of November 2022, the Surface Pressure Anomalies Pattern displayed a large high-pressure area over the North Atlantic, drawing cold Arctic and Siberian air over the continent. The presence of a low pressure over Central Europe further favored cold air advection. This resulted in Temperature Anomalies of up to 10∘ below average. The high-pressure area in the North Atlantic persisted until early December, after which warmer air masses from the North Atlantic spread over large parts of Europe.] Important: make sure to refer to both panels in the first row of maps and use the panel titles to refer to them, i.e. Surface Pressure Anomalies Pattern or Temperature Anomalies or wind speed or Precipitation.

## A4    Climate and Data Background for the Analysis

«According to the/In Chapter XX of the» IPCC AR6 report «brief summary of confidence level on change in frequency/intensity of the selected extreme» [example: it is virtually certain that there has been a decrease in severity and/or frequency of cold spells in the last several decades, and the consensus is that at a global level this decrease will continue in the future.]. «Additional information about specific location/event type if relevant» [example: In Scandinavia, cold spells have become on average 4 ℃ warmer since 1950]. Our analysis approach rests on looking for weather situations similar to those of the event of interest having been observed in the past. For «event name», we have:

    – «high confidence in the robustness of our approach given the available climate data, as the event is very similar to other past events in the data record [if right-hand-side dial points to the left]

    – medium-high confidence in the robustness of our approach given the available climate data, as the event is similar to other past events in the data record [if right-hand-side dial points ¾ to the left]

    – medium-low confidence in the robustness of our approach given the available climate data, as the event is unusual in the data record [if right-hand-side dial points ¾ to the right]

    – low confidence in the robustness of our approach given the available climate data, as the event is largely unique in the data record [if right-hand-side dial points to the right]» .



## A5    ClimaMeter Analysis

We analyse here (see methodology for more details) how events similar to «event name» have changed in the present (2001–2022) compared to what they would have looked like if they had occurred in the past (1979–2000). Surface Pressure Changes show «brief description of the changes here» [example: that the high pressure over the North Atlantic has become weaker than in the past, resulting in weaker cold-air advection over Europe]. «Temperature Changes/Precipitation Changes/Wind Changes» show that similar events produce «variable name here» which in the present climate are «brief description of the changes here» [example: temperatures which in the present climate are between 1 ℃ and 4 ℃ hotter than what they would have been in the past]. «This has resulted in/This coincided with/similar connective phrase» «description of variable changes over cities as shown in figure» than they would have been in the past [example: temperatures in Berlin and Stockholm having become 5 ℃ warmer than they would have been in the past]. We also note that Similar Past Events «description of seasonal changes in occurrence of the analogues» [example: have become more common in the spring than in the winter months, contributing to making the cold spells less severe.] «Finally, we find that sources of natural climate variability, notably the «Pacific Decadal Oscillation/El Nino—Southern Oscillation/Atlantic Multidecadal Oscillation», may have heavily influenced the event. This means that the changes we see in the event compared to the past may be primarily due to natural climate variability [If left-hand-side-dial points to the left]».

- «Finally, we find that sources of natural climate variability, notably the «Pacific Decadal Oscillation/El Nino—Southern Oscillation/Atlantic Multidecadal Oscillation», may have heavily influenced the event. This means that the changes we see in the event compared to the past may be primarily due to natural climate variability [If left-hand-side-dial points to the left]».

- «Finally, we find that sources of natural climate variability, notably the «Pacific Decadal Oscillation/El Nino—Southern Oscillation/Atlantic Multidecadal Oscillation», may have influenced the event. This suggests that the changes we see in the event compared to the past may be partly due to human driven climate change, with a contribution from natural variability. [If left-hand-side-dial points ¾ to the left]»

- «Finally, we find that sources of natural climate variability, notably the «Pacific Decadal Oscillation/El Nino—Southern Oscillation/Atlantic Multidecadal Oscillation», may have only partly influenced the event. This means that the changes we see in the event compared to the past may be mostly due to human driven climate change [If left-hand-side-dial points ¾ to the right]»

- «Finally, we find that sources of natural climate variability did not influence the event. This means that the changes we see in the event compared to the past may be primarily due to human driven climate change [If left-hand-side-dial points to the right]»



## A6   Conclusion

Based on the above, we conclude that «event type» similar to «event name» have become «short description of how the circulation change likely affected the intensity of the event». [example: have become 3 ℃ warmer than in the present than in the past]. We interpret «event name» as:

– a largely unique event [if right-hand-dial points to the right]

– an unusual event [if right-hand-dial points ¾ to the right]

– an event [if right-hand-dial points ¾ to the left or to the left]

– whose characteristics can be ascribed to human driven climate change [if arrow of left-hand-side-dial all the way to the right]

– whose characteristics can mostly be ascribed to human driven climate change [if arrow of left-hand-side-dial points ¾ to the right]

– for which natural climate variability played a role [if arrow of left-hand-side-dial points ¾ to the left].

– for which natural climate variability likely played an important role [if arrow of left-hand-side-dial all the way to the left].

*Author contributions.* DF performed the analyses, created the ClimaMeter figures and the ClimaMeter Website. GM created the template for reports. DF, GM, EC, TA, MV, FP, PY, and RV devised the methodology underlying ClimaMeter. DF, EC, TA, MV, FP, PY, MSL, AH, and PB contributed to the visualization. All authors contributed to write the paper and discussed the results.

*Competing interests.* The authors declare no competing interests.

*Disclaimer.* MSWX (Beck et al., 2022) data are freely available at https://www.gloh2o.org/mswx/.

*Acknowledgements.* The authors thank Bérengère Dubrulle for useful suggestions. The authors acknowledge the support of the INSU-CNRS-LEFE-MANU grant (project CROIRE), the grant ANR-20-CE01-0008-01 (SAMPRACE) and support from the European Union's Horizon 2020 research and innovation programme under grant agreement No. 101003469 (XAIDA), and the Marie Sklodowska-Curie grant agreement No. 956396 (EDIPI).



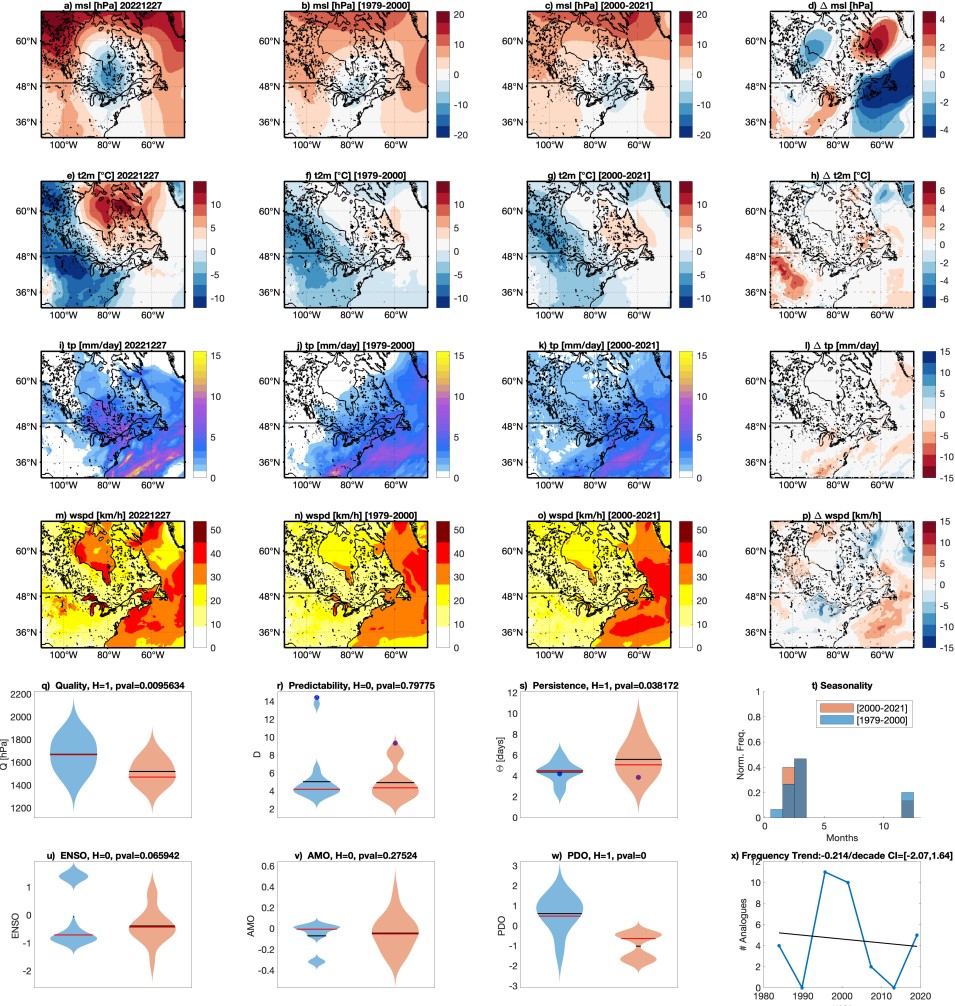

**Figure A1.** 2022/12/21-26 North American Winter Storm. Average surface pressure anomalies (msl) (a), average 2-meter temperatures anomalies (t2m) (e), cumulated total precipitation (tp) (i), and average wind speed (wspd) in the period of the event. Average of the surface pressure analogues found in the counterfactual [1979–2000] (b) and factual periods [2001–2022] (c), along with corresponding 2-meter temperatures (f, g), cumulated precipitation (j, k), and wind speed (n, o). Changes between present and past analogues are presented for surface pressure $\Delta$slp (d), 2 meter temperatures $\Delta$t2m (h), total precipitation $\Delta$tp (i), and wind speed $\Delta$wspd (p): color-filled areas indicate significant anomalies with respect to the bootstrap procedure. Violin plots for past (blue) and present (orange) periods for Quality of analogues $Q$ (q), Predictability Index $D$ (r), Persistence Index $\Theta$ (s), and distribution of analogues in each month (t). Violin plots for past (blue) and present (orange) periods for ENSO (u), AMO (v) and PDO (w). Number of the Analogues occurring in each subperiod (blue) and linear trend (black). Horizontal bars in panels (q,r,s,u,v,w) correspond to the mean (black) and median (red) of the distributions. Values for the peak day of the extreme event are marked by a blue dot.



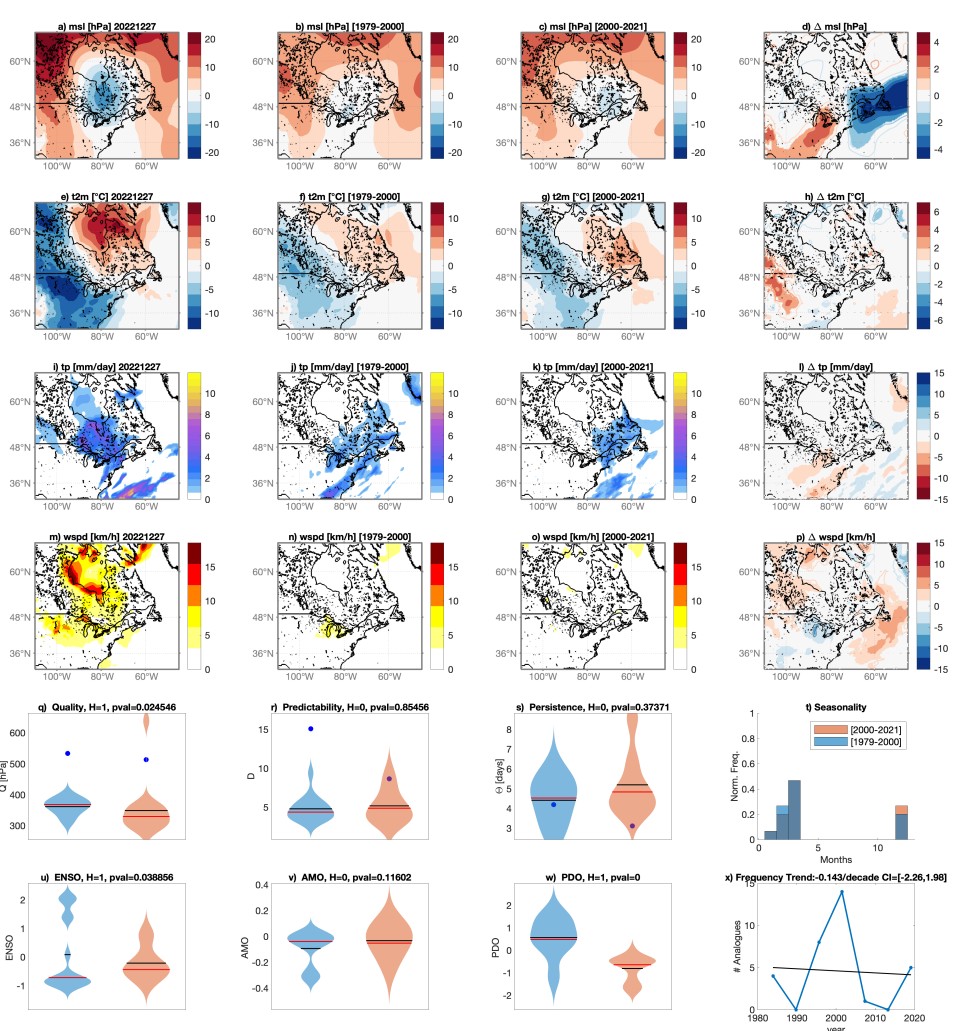

**Figure A2.** 2022/12/21-26 North American Winter Storm. As in Figure A1 but for ERA5 data



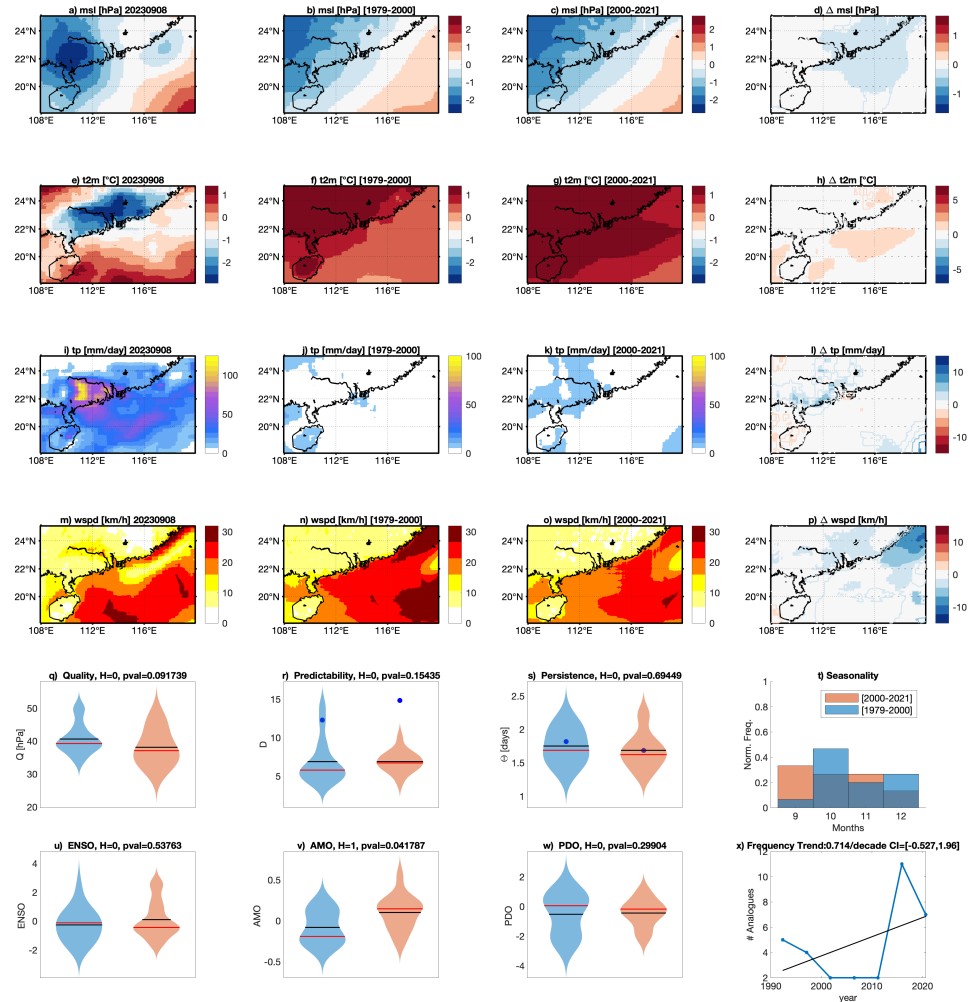

**Figure A3.** 2023/09/08 Guangdong - Hong Kong Floods. Average of surface pressure anomalies (msl) (a), average 2-meter temperatures anomalies (t2m) (e), cumulated total precipitation (tp) (i), and average wind speed (wspd) in the period of the event. Average of the surface pressure analogues found in the counterfactual [1979–2000] (b) and factual periods [2001–2022] (c), along with corresponding 2-meter temperatures (f, g), cumulated precipitation (j, k), and wind speed (n, o). Changes between present and past analogues are presented for surface pressure $\Delta$slp (d), 2 meter temperatures $\Delta$t2m (h), total precipitation $\Delta$tp (i), and wind speed $\Delta$wspd (p): color-filled areas indicate significant anomalies with respect to the bootstrap procedure. Violin plots for past (blue) and present (orange) periods for Quality of analogues $Q$ (q), Predictability Index $D$ (r), Persistence Index $\Theta$ (s), and distribution of analogues in each month (t). Violin plots for past (blue) and present (orange) periods for ENSO (u), AMO (v) and PDO (w). Number of the Analogues occurring in each subperiod (blue) and linear trend (black). Horizontal bars in panels (q,r,s,u,v,w) correspond to the mean (black) and median (red) of the distributions. Values for the peak day of the extreme event are marked by a blue dot.





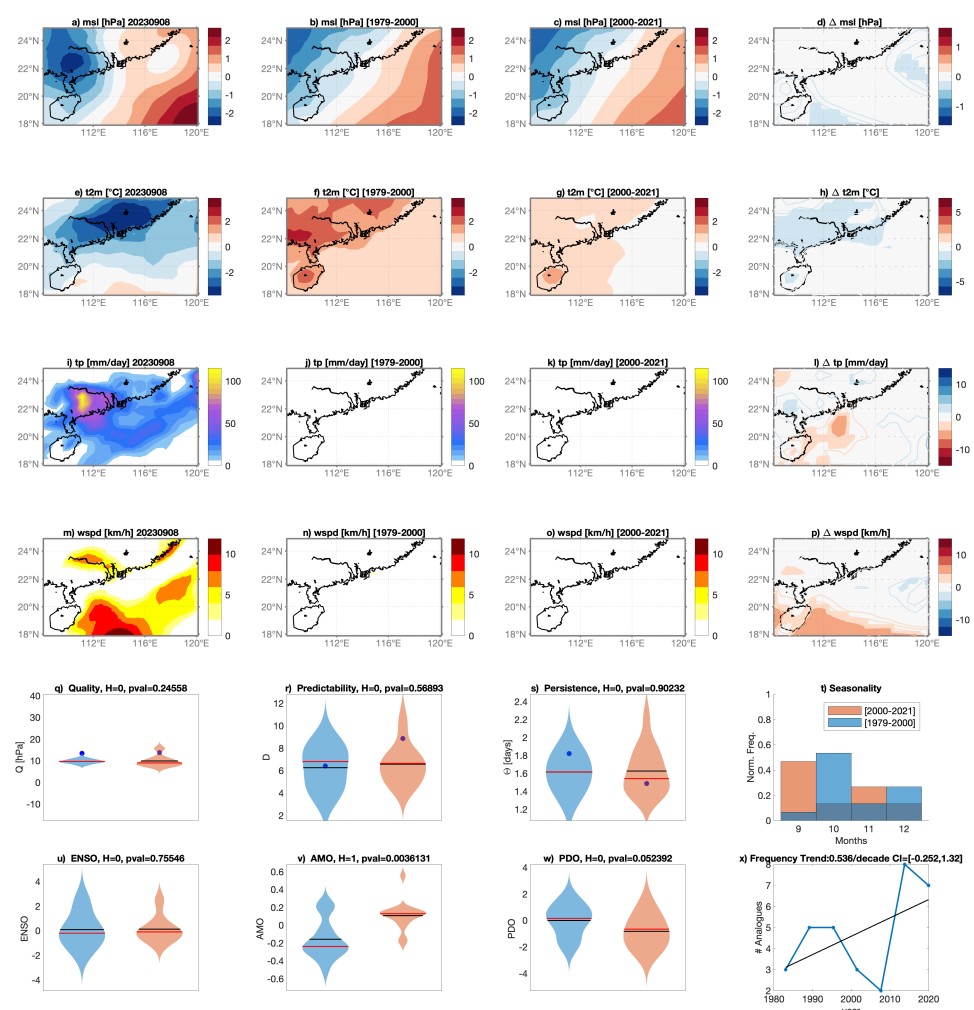

**Figure A4.** 2023/09/08 Guangdong - Hong Kong Floods. As in Figure A3 but for ERA5 data.



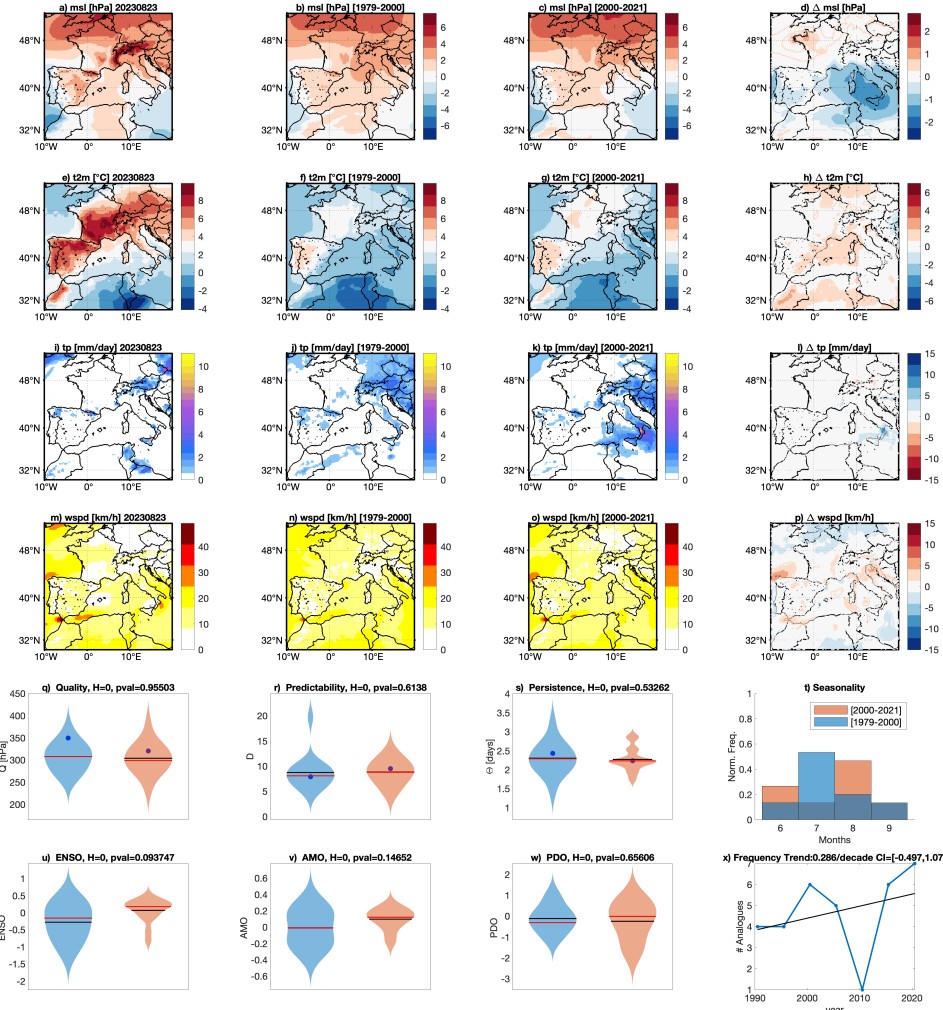

**Figure A5.** 2023/08/20-23 Late Summer French Heatwave. Average of surface pressure anomalies (msl) (a), average 2-meter temperatures anomalies (t2m) (e), cumulated total precipitation (tp) (i), and average wind speed (wspd) in the period of the event. Average of the surface pressure analogues found in the counterfactual [1979–2000] (b) and factual periods [2001–2022] (c), along with corresponding 2-meter temperatures (f, g), cumulated precipitation (j, k), and wind speed (n, o). Changes between present and past analogues are presented for surface pressure $\Delta$slp (d), 2 meter temperatures $\Delta$t2m (h), total precipitation $\Delta$tp (i), and wind speed $\Delta$wspd (p): color-filled areas indicate significant anomalies with respect to the bootstrap procedure. Violin plots for past (blue) and present (orange) periods for Quality of analogues $Q$ (q), Predictability Index D (r), Persistence Index $\Theta$ (s), and distribution of analogues in each month (t). Violin plots for past (blue) and present (orange) periods for ENSO (u), AMO (v) and PDO (w). Number of the Analogues occurring in each subperiod (blue) and linear trend (black). Horizontal bars in panels (q,r,s,u,v,w) correspond to the mean (black) and median (red) of the distributions. Values for the peak day of the extreme event are marked by a blue dot.



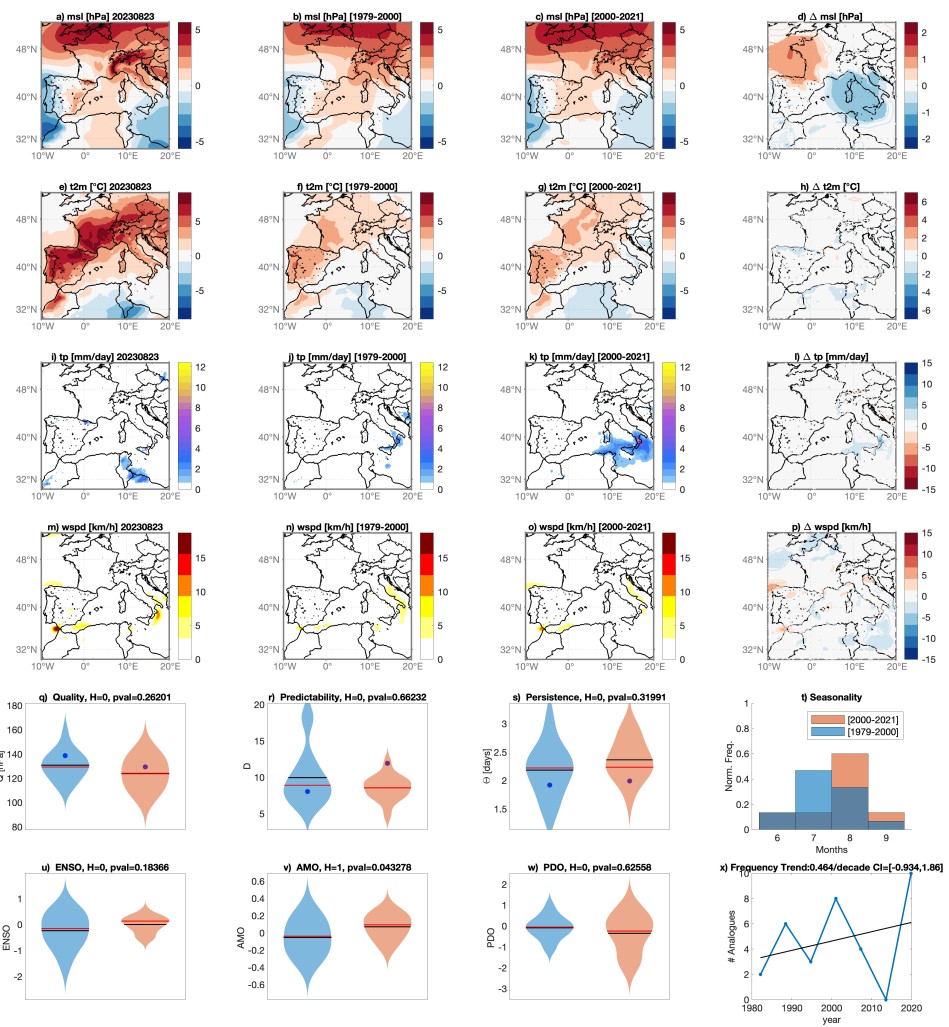

**Figure A6.** 2023/08/20-23 Late Summer French Heatwave. As in Figure A5 but for ERA5 data.





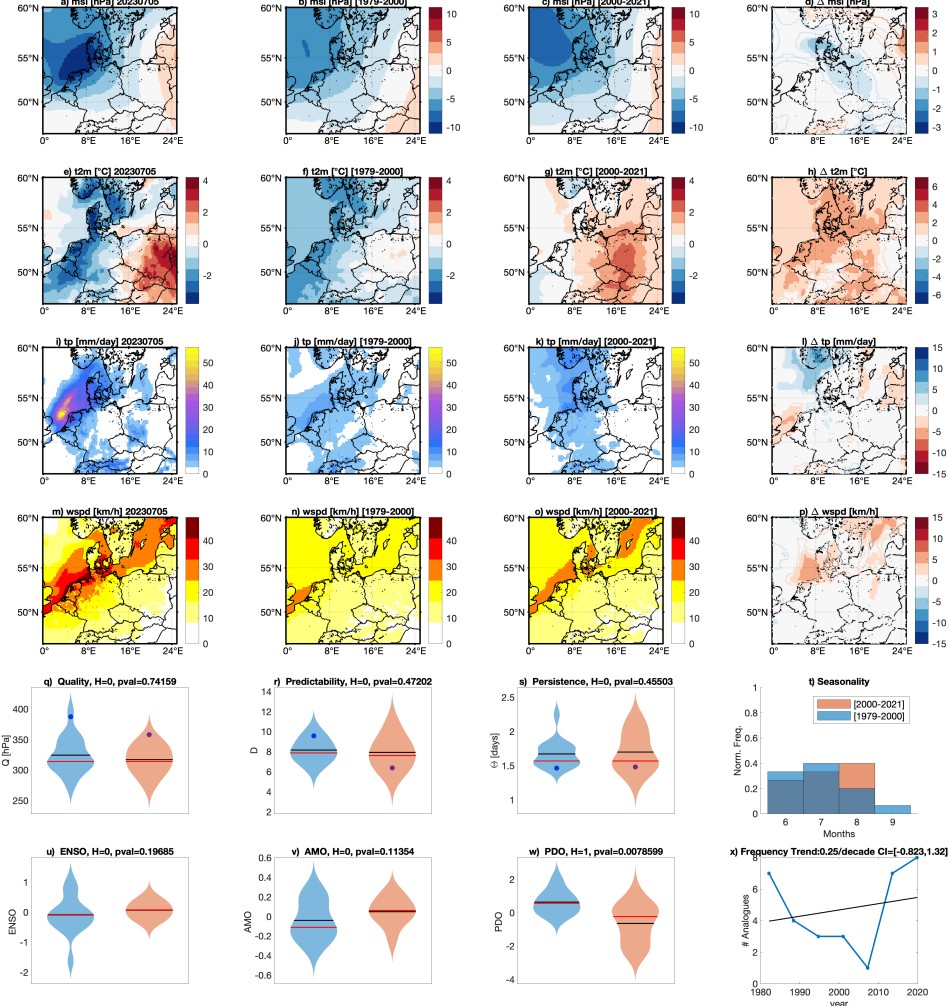

**Figure A7.** 2023/07/05 - Storm Poly in Northern Europe. Surface pressure anomalies (msl) (a), average 2-meter temperatures anomalies (t2m) (e), cumulated total precipitation (tp) (i), and average wind speed (wspd) in the period of the event. Average of the surface pressure analogues found in the counterfactual [1979–2000] (b) and factual periods [2001–2022] (c), along with corresponding 2-meter temperatures (f, g), cumulated precipitation (j, k), and wind speed (n, o). Changes between present and past analogues are presented for surface pressure $\Delta$slp (d), 2 meter temperatures $\Delta$t2m (h), total precipitation $\Delta$tp (i), and wind speed $\Delta$wspd (p): color-filled areas indicate significant anomalies with respect to the bootstrap procedure. Violin plots for past (blue) and present (orange) periods for Quality of analogues $Q$ (q), Predictability Index D (r), Persistence Index $\Theta$ (s), and distribution of analogues in each month (t). Violin plots for past (blue) and present (orange) periods for ENSO (u), AMO (v) and PDO (w). Number of the Analogues occurring in each subperiod (blue) and linear trend (black). Horizontal bars in panels (q,r,s,u,v,w) correspond to the mean (black) and median (red) of the distributions. Values for the peak day of the extreme event are marked by a blue dot.



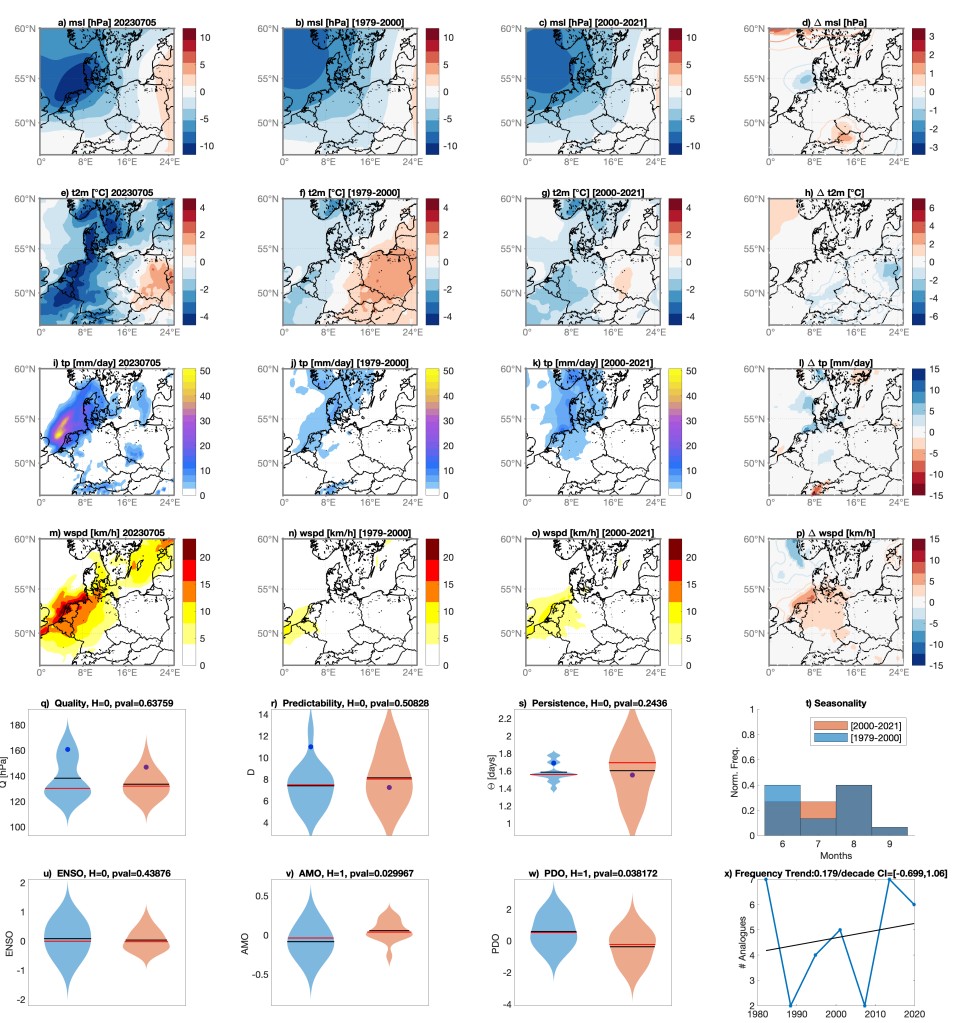

**Figure A8.** 2023/07/05 - Storm Poly in Northern Europe. As in Figure A7 but for ERA5 data.



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
