# Peer review of "ClimaMeter: Contextualising Extreme Weather in a Changing Climate"

_EGUsphere, 2023_

## Referee Comment (RC1)

**EGUsphere-2023-2643 reviewer comments**

This paper documents the ClimaMeter online platform in detail, outlining the methodology used to carry out the analysis but also describing the graphical elements used to visualise the results, and the protocol used to write the reports. Examples of previous reports are included, with more detailed analysis provided in the supplementary material (although these are not discussed in the main text), along with templates for writing the reports.

The paper is well written throughout, and the methodology and protocol are clearly defined. However, it is not entirely clear to me what the purpose of the paper is: is the intention merely to publish the protocol template to allow readers to produce their own ClimaMeter-style analyses, or to establish the scientific basis for the project? The level of detail sometimes suggests the former, but I think that the paper would benefit from offering more insight into the reasoning behind the modelling and reporting choices made, both to provide the scientific foundations for the work, and also as a guide to understanding and interpreting the official ClimaMeter output.

Overall the ClimaMeter approach is a valuable addition to the D&A toolkit. I anticipate that the paper will be suitable for publication after relatively minor changes: no new analysis is required, but more discussion of the analyses presented and reflection on the rationale behind some of the modelling choices is needed.

**General issues**

My main criticism of this paper would be that some of the most interesting information has been relegated to figures in the appendices. The reports reproduced in section 5 are, in essence, publicly available elsewhere: rather than reproducing them here, it would be useful to see more detailed discussion of the elements of the additional plots, particularly the diagnostic plots showing analogue quality, predictability and persistence, and an explanation of how those plots should be interpreted. This should be done for at least one of the studies in detail, and preferably for all (although in this case I would just highlight the most interesting features).

Similarly, there needs to be some discussion in the main text of how much the results might be altered if using the ERA5 dataset, rather than MSWX: and if the results are very different, is this because of the weather conditions on the analogue days, or because entirely different analogues are found? Are the results more stable for some hazards than others? An understanding of these potential limitations of the method is vital to understand when the method will be of most use, and when other methods may be more appropriate.

**Specific points**

40-44. Storyline-based (or reforecast-based) approaches to attribution do consider extreme weather events in terms of the weather system: while I appreciate the distinction between that and the

ClimaMeter approach, the existence of such methods should at least be acknowledged here. Also suggest changing 'classical' to 'probabilistic' to highlight exactly what is meant by the term.

73-76. A more thorough discussion of potential limitations of the method is needed. How sensitive is the analogue quality to the domain used, or to the choice of dataset? How sensitive are the results of the analysis to these factors? You could also add that, while there is a risk of underestimating the role of climate change due to the warming state of the climate during the reference period, the comparison is at least well defined and avoids extrapolation beyond the available data.

77. 'Data download and pre-processing' doesn't quite cover all of the steps in this section - perhaps 'data pre-processing and analysis'?

87-88. If MSWX did provide mid-tropospheric fields, would it be preferable to use those? Has any testing been done to understand how much difference this would make to the results?

95. The method actually used to find the analogues gets a bit lost here - I'd suggest moving 'that is, the surface pressure fields minimizing the Euclidean distance to the event itself' after 'in terms of the event's surface pressure pattern only' so that the analogues are defined right away. Similarly, 'once analogues are found… best 15 in each period' seems to belong at the start of point 5.

140-145. First, please clarify in the text how 'the dial points 95% to the right' should be interpreted (for readers unfamiliar with the format it's confusing to keep having to refer to the plot to check). Secondly, It should be highlighted that the relative sizes of the effects of climate change and natural variability are not actually estimated, which could be considered a limitation of the method when compared to the standard probabilistic approach.

Finally, I think the reasoning behind this dial needs to be explained in more detail. The argument here is that if the 15 analogues in the past are consistently in a different phase to the 15 analogues in the present, then this constitutes evidence that natural variability contributed significantly to the difference from past to present. This seems (partly) counter-intuitive to me, in that I would expect some circulation patterns to be more likely to appear in particular phases of ENSO in both periods and the phases of ENSO to be independent of the period chosen: so I wouldn't expect the best analogues in the past to be systematically associated with a different phase of ENSO (I can see the argument more clearly in the case of lower-frequency modes such as the AMO, where each period may be dominated by a different phase). Can you elaborate on this? Or is it actually the case that ENSO is less often found to have a significant effect? This is perhaps something that could be discussed with reference to Figure 2 (I note that it's very unusual for all three modes of variability to have a significant effect, but can only speculate as to why).

148: Suggest rephrasing as 'Q is simply the average Euclidean distance of each analogue from all other analogues' or similar. Otherwise this almost seems to suggest that 15 more analogues are found for each of the analogues, and Q computed for those.

149-155. Which category is assigned if Q is below the 75th percentile in one period and between the 75th and 95th in the other? (it may be easier to rephrase these categories in terms of 'below the

95th percentile' to ensure an exhaustive definition, and to capture any cases where one period is above the 95th and the other is below the 75th?)

170. Why are the significance maps only included for surface pressure changes but not the hazard variable? I would have thought (perhaps naively) that significant changes in the hazard would be of more interest.

173. Please add a line explaining why the Cramer-von Mises test was used (presumably to compare the two distributions nonparametrically - is this a more stringent test than just comparing means, and therefore more likely to assign differences to indices of natural variability than to climate change?)

179-181. The choice to stick to a pre-specified report format may seem like an odd one to many scientists reading this paper, so I think it would be worth adding a line or two here explaining the benefits (and limitations) of doing so. It would be particularly interesting to hear of any feedback from the media on this - do journalists find it easier to digest this kind of complex information when it is presented in a format that they have become familiar with?

199. 'As soon as possible' can mean very different things, so I think it would be useful to highlight just how quickly this analysis can be produced - please add the typical/target timeframe for release.

201. What kind of feedback prompts an update of the report?

205-223. I don't think this description of the website structure adds anything to the paper, and would recommend removing it. However, a discussion of the details behind Figure 2 (perhaps just before the conclusions) could be informative - for example, discussing the relative frequency with which each mode of variability is found to be significant, and commenting on the fact that a high proportion of events studied have no close analogues: is this because the core team are choosing to study the more meteorologically unusual events, so we shouldn't expect to find any close analogues? Or is there some factor that could be affecting the quality of the analogues identified, such as sensitivity to the domain size?

224-399. It's not clear to me what the purpose is of simply reproducing these four studies here without any further discussion or analysis. For readers hoping to replicate the analysis for another event, it would be far more useful to choose one study to focus on, and to walk through the process of defining the event and interpreting the analysis. The more detailed figure for that study should be moved from the Appendix to the main text, and the elements that are not already discussed in the text commented on  (in particular, the violinplots showing the predictability and persistence, and the plots of trends in the distribution of analogues). The differences and similarities between the MSWX and ERA5 results should also be addressed here. If the four separate case studies are retained in the main text, there needs to be some discussion in the conclusion of what they illustrate about the method.

231. Broken Wikipedia link/citation.

282. Should be 'Haikui'.

402. Typo: 'this critical issue'.

401-404. This paragraph rather implies that no other methods or groups exist to communicate the changing risks of extreme events, which is simply not true (there are now several operational met services carrying out rapid attribution studies and communicating them to the media, as well as WWA). It would be more accurate to say that ClimaMeter is *part of* a continuing effort to contextualise extreme and hazardous events: to me, the innovation here is that, rather than using statistical methods that analyse time series of events that may arise from different processes, ClimaMeter considers changes in extreme weather arising from the same circulation patterns, which allows a more nuanced understanding of the spatial and multivariate changes associated with the weather type of interest. (This is particularly important for wind and precipitation, because unlike in univariate probabilistic attribution,  the event definition doesn't require averaging over the spatial domain and so doesn't smooth out the local extremes)

410. Rather than referring to the four case studies, it would be more relevant to refer to the map in Figure 2.

411-412. I'm not sure what 'These underscore the significance of contextualising extreme events, as a tool to understand the broader context within which they occur' means here.

490. This should be updated to 2001-2024 to be current at time of publication.

Figures A1-A8. The values of ENSO, AMO and PDO associated with the event are missing from the plots.

---

## Referee Comment (RC2)

Review for egusphere-2023-2643:

"ClimaMeter: Contextualising Extreme Weather in a Changing Climate"

This manuscript presents a novel platform for assessing extreme events ("ClimaMeter"), which offers a useful database for past and future analyses. The authors present the protocol for ClimaMeter and explain its output and how the events are visualised. The presentation is clear and well-structured. In my opinion, the manuscript should be acceptable for publication after minor revisions.

**General comments**

In addition to reviewing the manuscript, I have read the report of Referee #1 and generally agree with their comments. In particular, I had the same confusion about the exact purpose of the paper. The discussion is very general, with explanations that could be reported on the ClimaMeter website, and does not have the specific details you would expect from a scientific paper. A more detailed description of (at least) one of the studies, as Referee #1 proposed, would certainly improve the scope of the paper.

**Specific comments**

- L53-55 – A bit more detail could be given on the event selection. Are there any plans on making this a more automatic process?

- L72 – A climate reconstruction still uses a combination of observations and models; how is this independent of model bias?

- L81 – It would be useful to report the actual grid size of MSWX here.

- L94 – It is unclear to me how these analogues are exactly defined. Do you ever consider similar events at other locations?

- L140-145 – The way the gauge is presented makes it seem that the conclusion of natural variability vs climate change is very certain, but it is based on only three indices. You mention some of the drawbacks of this earlier in the text, but the final presentation (i.e. the left-hand gauge) can lead to the interpretation that an event is (for example) completely outside of natural variability, while there may be other factors playing a role. A more detailed description of the reasoning behind these choices will be helpful.

**Technical corrections**

- L17 – "Hurrícane" → "Hurricane"
- L113 – "different" → "difference"
- L136 – "or wind" → "and wind" (as you show them all)
- L137 – ""past" and "present"" → ""present" and "past"" (as you show "present" — "past")
- Fig. 1 – ""past" and "present"" → ""present" and "past"" (as you show "present" — "past")
- L163 – "or wind" → "and wind" (as you show them all)
- L209 – "analyzed" → "analysed"

---

## Author Response (AR1)

Dear Editor,

We have carefully considered the comments provided by both Anonymous Referee #1 and #2 and have made significant revisions to address their suggestions and improve the clarity and scientific robustness of our paper.

In response to the specific comments raised by the referees, we have made several revisions. To address the concern about relegating important information to figures in the appendices, we have limited the number of example reports provided in the main text to two and included additional plots with detailed commentary. Furthermore, we have expanded the discussion of the differences between the MSWX and ERA5 datasets and provided a more thorough examination of the potential limitations of our method, including sensitivity to domain and dataset choices.

We have also clarified the approach used to identify analogues, ensuring that this key part of our methodology is easily reproducible by readers. Additionally, we have provided a more detailed explanation of the rationale behind the dial used to assess the relative sizes of the effects of climate change and natural variability.

Furthermore, we have revised the text to provide more context and explanation for certain aspects of our methodology and findings. We have addressed technical corrections and improved the overall coherence and readability of the manuscript.

In summary, we believe that the revisions made in response to the reviewers' comments have significantly strengthened the manuscript. We are confident that the revised version of the paper represents a valuable contribution to the field of weather and climate dynamics, and we hope that it will be suitable for publication in your esteemed journal.

Thank you once again for the opportunity to submit our work to Weather and Climate Dynamics, and we look forward to hearing from you regarding the status of our manuscript.

Sincerely,

Davide Faranda
on behalf of the ClimaMeter team

**Anonymous Referee #1**

This paper documents the ClimaMeter online platform in detail, outlining the methodology used to carry out the analysis but also describing the graphical elements used to visualize the results, and the protocol used to write the reports. Examples of previous reports are included, with more detailed analysis provided in the supplementary material (although these are not discussed in the main text), along with templates for writing the reports.

The paper is well written throughout, and the methodology and protocol are clearly defined. However, it is not entirely clear to me what the purpose of the paper is: is the intention merely to publish the protocol template to allow readers to produce their own ClimaMeter-style analyses, or to establish the scientific basis for the project? The level of detail sometimes suggests the former, but I think that the paper would benefit from offering more insight into the reasoning behind the modelling and reporting choices made, both to provide the scientific foundations for the work, and also as a guide to understanding and interpreting the official ClimaMeter output.

Overall the ClimaMeter approach is a valuable addition to the D&A toolkit. I anticipate that the paper will be suitable for publication after relatively minor changes: no new analysis is required, but more discussion of the analyses presented and reflection on the rationale behind some of the modelling choices is needed.

**We thank the Reviewer for taking the time to review our paper on the ClimaMeter online platform. We appreciate their positive feedback on the clarity of our methodology, protocol, and documentation of the graphical elements used in visualizing the results, as well as the suggestions for improvement. Regarding the purpose of the paper, our intention is twofold: firstly, to provide a comprehensive overview of the ClimaMeter platform, including its methodology and protocol, to enable readers to replicate our analyses. Secondly, we aim to provide a scientific basis for the project. To ensure that this second goal is achieved, we will follow the Reviewer's suggestion of including a more detailed discussion of the methodological choices in the revised version of the manuscript. We provide further detail on this point in our replies to the Reviewer's specific comments below.**

My main criticism of this paper would be that some of the most interesting information has been relegated to figures in the appendices. The reports reproduced in section 5 are, in essence, publicly available elsewhere: rather than reproducing them here, it would be useful to see more detailed discussion of the elements of the additional plots, particularly the diagnostic plots showing analogue quality, predictability and persistence, and an explanation of how those plots should be interpreted. This should be done for at least one of the studies in detail, and preferably for all (although in this case I would just highlight the most interesting features).

Similarly, there needs to be some discussion in the main text of how much the results might be altered if using the ERA5 dataset, rather than MSWX: and if the results are very different, is this because of the weather conditions on the analogue days, or because entirely different analogues are found? Are the results more stable for some hazards than others? An understanding of these

potential limitations of the method is vital to understand when the method will be of most use, and when other methods may be more appropriate.

**We recognize the importance of discussing key information within the main text rather than relegating it to appendices. Following the Reviewer's suggestion, we have decided to limit the number of example reports provided in the main text to two, and include additional plots for those. The other two examples have been removed. In particular, we have expanded the discussion of the differences between the MSWX and ERA5 data, and included the ERA5 figures in the main paper instead. We will also provide a summary of our qualitative evaluation of the differences between the two datasets for different hazard types. However, we see no easy way of including a robust quantitative evaluation of this in the paper, as we do not currently have enough ClimaMeter reports to account for seasonal and geographical dependence of such differences for multiple hazard types. We have mentioned this point as a limitation of our current approach, and something that should be investigated once we have processed a larger number of extreme events.**

**The additional plots included in the main text are commented in detail, and we have also updated the text describing the different evaluation metrics in Sect. 2.**

**The above edits will hopefully improve the clarity of our methodology, as well as provide a more robust scientific basis for the whole ClimaMeter protocol, in response to the Reviewer's overarching comment.**

*Specific points*

40-44. Storyline-based (or reforecast-based) approaches to attribution do consider extreme weather events in terms of the weather system: while I appreciate the distinction between that and the ClimaMeter approach, the existence of such methods should at least be acknowledged here. Also suggest changing 'classical' to 'probabilistic' to highlight exactly what is meant by the term.

**We agree with the Reviewer and will update this passage to cite work such as van Garderen (2020) and Leach et al. (2021), which are good examples of storyline-based and forecast-based attribution approaches (although we are aware that these are only two of many studies on these topics). Your suggestion to replace "classical" with "probabilistic" to clarify the intended meaning of the term is well-taken, and we will update the text accordingly.**

**van Garderen, L., Feser, F., and Shepherd, T. G.: A methodology for attributing the role of climate change in extreme events: a global spectrally nudged storyline, Nat. Hazards Earth Syst. Sci., 21, 171–186, https://doi.org/10.5194/nhess-21-171-2021, 2021.**

**Leach, N. J., Weisheimer, A., Allen, M. R., & Palmer, T. (2021). Forecast-based attribution of a winter heatwave within the limit of predictability. *Proceedings of the National Academy of Sciences*, *118*(49), e2112087118.**

**Jun Wang et al. ,Storyline attribution of human influence on a record-breaking spatially compounding flood-heat event.Sci. Adv.9,eadi2714(2023).DOI:10.1126/sciadv.adi271**

73-76. A more thorough discussion of potential limitations of the method is needed. How sensitive is the analogue quality to the domain used, or to the choice of dataset? How sensitive are the results of the analysis to these factors? You could also add that, while there is a risk of underestimating the role of climate change due to the warming state of the climate during the reference period, the comparison is at least well defined and avoids extrapolation beyond the available data.

**Thank you for your feedback. We agree on the need to expand on the method's potential limitations, including sensitivity to domain and dataset choices (for the latter, see also our reply to the Reviewer's previous comment regarding the use of ERA5). We also agree that our approach is very conservative, in that our reference (past) period is not a preindustrial climate, but rather a mid-to-late 20th century climate. We have added a discussion of these limitations in Sect. 2, and also mention them in the conclusions..**

77. 'Data download and pre-processing' doesn't quite cover all of the steps in this section - perhaps 'data pre-processing and analysis'?

**We agree, and have updated the subsection title as suggested.**

87-88. If MSWX did provide mid-tropospheric fields, would it be preferable to use those? Has any testing been done to understand how much difference this would make to the results?

**Thank you for raising these important points. Indeed, previous research from some of the authors of this study (Fery and Faranda, 2023; Jezequel et al., 2018) has emphasized the importance of geopotential height, especially for deep convective events and heatwaves. Moreover, Holmberg et al. (2023) showed that analogues selected on Z500 can yield regionally different results from those selected on SLP. However, the results in Faranda et al. (2022) have shown that surface fields are nonetheless reasonably effective for detecting analogues of extreme events. In conjunction with the new discussion of the limitations of our approach (see our reply to a previous comment), we discuss the rationale for using surface fields. We also present the analogues computed using Z500 from ERA5 rather than SLP for the two examples discussed in the main text.**

**Fery, L., & Faranda, D. (2024). Analysing 23 years of warm-season derechos in France: a climatology and investigation of synoptic and environmental changes. *Weather and Climate Dynamics, 5(1), 439-461.***

**Holmberg, E., Messori, G., Caballero, R., & Faranda, D. (2023). The link between European warm-temperature extremes and atmospheric persistence. *Earth System Dynamics*, *14*(4), 737-765.**

**Jézéquel, A., Yiou, P., and Radanovics, S.: Role of circulation in European heatwaves using flow analogues, Clim. Dynam., 50, 1145–1159, https://doi.org/10.1007/s00382-017-3667-0, 2018. a**

95. The method actually used to find the analogues gets a bit lost here - I'd suggest moving 'that is, the surface pressure fields minimizing the Euclidean distance to the event itself' after 'in terms of

the event's surface pressure pattern only' so that the analogues are defined right away. Similarly, 'once analogues are found… best 15 in each period' seems to belong at the start of point 5.

**In our revised manuscript, we have followed the Reviewer's suggestion to clarify the approach used to identify the analogues. This is a key part of our methodology and needs to be easily reproducible by the readers.**

140-145. First, please clarify in the text how 'the dial points 95% to the right' should be interpreted (for readers unfamiliar with the format it's confusing to keep having to refer to the plot to check). Secondly, It should be highlighted that the relative sizes of the effects of climate change and natural variability are not actually estimated, which could be considered a limitation of the method when compared to the standard probabilistic approach.

**We have clarified the interpretation of the dial in the revised text. Concerning the Reviewer's second point, we highlight in the new discussion of methodological limitations that we indeed do not estimate quantitatively the relative sizes of the effects of climate change and natural variability, contrary to other probabilistic approaches. This will hopefully provide readers with a comprehensive understanding of the method's capabilities and constraints.**

Finally, I think the reasoning behind this dial needs to be explained in more detail. The argument here is that if the 15 analogues in the past are consistently in a different phase to the 15 analogues in the present, then this constitutes evidence that natural variability contributed significantly to the difference from past to present. This seems (partly) counter-intuitive to me, in that I would expect some circulation patterns to be more likely to appear in particular phases of ENSO in both periods and the phases of ENSO to be independent of the period chosen: so I wouldn't expect the best analogues in the past to be systematically associated with a different phase of ENSO (I can see the argument more clearly in the case of lower-frequency modes such as the AMO, where each period may be dominated by a different phase). Can you elaborate on this? Or is it actually the case that ENSO is less often found to have a significant effect? This is perhaps something that could be discussed with reference to Figure 2 (I note that it's very unusual for all three modes of variability to have a significant effect, but can only speculate as to why).

**We understand this concern. In our revised manuscript, we provide a more detailed explanation of the rationale behind the dial and its interpretation. As the Reviewer states, specific circulation patterns may appear more often in conjunction with specific phases of a given large-scale variability mode. If, in both periods, this is true (continuing the Reviewer's example, if all analogues in both periods appear during El Nino), then we make the (heavily simplified) assumption that ENSO did non contribute to the differences we see - i.e. no role of natural variability as associated with ENSO. If, however, analogues in the two periods occur during different ENSO phases, then this difference may affect the differences we see in the surface variables that we analyse. As the Reviewer correctly states, a priori one would not expect the best analogues in the past to be systematically associated with a different phase of ENSO than the analogues in the present, but empirically this occurs relatively often.  We counted how often each mode of variability presents a significant change between past and present analogues in all**

**the ClimaMeter reports we have produced to date, and will include a discussion of this statistic in the revised text (figure 2).**

148: Suggest rephrasing as 'Q is simply the average Euclidean distance of each analogue from all other analogues' or similar. Otherwise this almost seems to suggest that 15 more analogues are found for each of the analogues, and Q computed for those.

**We rephrased this passage, to clarify that Q is computed based on the distances between all analogues.**

149-155. Which category is assigned if Q is below the 75th percentile in one period and between the 75th and 95th in the other? (it may be easier to rephrase these categories in terms of 'below the 95th percentile' to ensure an exhaustive definition, and to capture any cases where one period is above the 95th and the other is below the 75th?)

**We agree that the current definition is ambiguous and will update this passage in the revised text. Specifically, the case described by the reviewer corresponds to the dial set at 65%.**

170. Why are the significance maps only included for surface pressure changes but not the hazard variable? I would have thought (perhaps naively) that significant changes in the hazard would be of more interest.

**We focus on the surface map to identify atmospheric circulation changes and then we use hazard variables to see the effects of these changes on precipitation, temperature, and wind speed. Furthermore, the surface pressure, being spatially smoother than hazard variables, is the most suitable field for a proper evaluation of Euclidean distances. Indeed also the hazard variables only show significant changes, there is an error in our text but not in the figures. This will be correct in the next version**

173. Please add a line explaining why the Cramer-von Mises test was used (presumably to compare the two distributions nonparametrically - is this a more stringent test than just comparing means, and therefore more likely to assign differences to indices of natural variability than to climate change?)

**We will add this explanation. As the Reviewer correctly notes, we selected the two-sided Cramer-von Mises test to compare pairs of distributions in a non parametric way, rather than using a parametric test for the mean.**

179-181. The choice to stick to a pre-specified report format may seem like an odd one to many scientists reading this paper, so I think it would be worth adding a line or two here explaining the benefits (and limitations) of doing so. It would be particularly interesting to hear of any feedback from the media on this - do journalists find it easier to digest this kind of complex information when it is presented in a format that they have become familiar with?

**We thank the Reviewer for this interesting suggestion. The Climameter figure has been indeed constructed taking into account the feedback received by journalists and colleagues working specifically in climate change communication. This is now reported in the text.**

199. 'As soon as possible' can mean very different things, so I think it would be useful to highlight just how quickly this analysis can be produced - please add the typical/target timeframe for release.

**Generally, we run analysis and produce the corresponding report within 2-3 days of the event. We specify this in the revised text.**

201. What kind of feedback prompts an update of the report?

**Feedback prompting an update of the report typically includes suggestions or criticisms that we receive.. So far, examples include colleagues or journalists finding errors in the figures of analysis, requests for clarification, updated estimates of the damage or of the meteorological values. Overall, we take into account any feedback that helps improve the reports and try to incorporate them as much as possible. This discussion is included in the updated version of the paper**

205-223. I don't think this description of the website structure adds anything to the paper, and would recommend removing it. However, a discussion of the details behind Figure 2 (perhaps just before the conclusions) could be informative - for example, discussing the relative frequency with which each mode of variability is found to be significant, and commenting on the fact that a high proportion of events studied have no close analogues: is this because the core team are choosing to study the more meteorologically unusual events, so we shouldn't expect to find any close analogues? Or is there some factor that could be affecting the quality of the analogues identified, such as sensitivity to the domain size?

**We understand the concern regarding the description of the website structure (sections 205-223), and we have moved the description to the introduction and shortened it. Regarding the event dashboard figure (Figure 3 in the current version of the manuscript), we have added a new figure (Figure 2 in the current version of the manuscript) to compute and discuss some statistics on the different modes of variability as per our reply to one of the previous comments. We now further clarify that, as the Reviewer states, that the generally poor quality of the analogues is due to the fact that we are selecting extreme events, whose large-scale meteorological drivers are often as unusual as the event itself.**

224-399. It's not clear to me what the purpose is of simply reproducing these four studies here without any further discussion or analysis. For readers hoping to replicate the analysis for another event, it would be far more useful to choose one study to focus on, and to walk through the process of defining the event and interpreting the analysis. The more detailed figure for that study should be moved from the Appendix to the main text, and the elements that are not already discussed in the text commented on (in particular, the violinplots showing the predictability and persistence, and the plots of trends in the distribution of analogues). The differences and similarities between the MSWX

and ERA5 results should also be addressed here. If the four separate case studies are retained in the main text, there needs to be some discussion in the conclusion of what they illustrate about the method.

**As per our previous replies, we now limit ourselves to only two case studies, but provide more detail and additional figures for these.**

231. Broken Wikipedia link/citation.

**No longer present in the text.**

282. Should be 'Haikui'.

**No longer present in the text.**

402. Typo: 'this critical issue'.

**We will change accordingly.**

401-404. This paragraph rather implies that no other methods or groups exist to communicate the changing risks of extreme events, which is simply not true (there are now several operational met services carrying out rapid attribution studies and communicating them to the media, as well as WWA). It would be more accurate to say that ClimaMeter is *part of* a continuing effort to contextualise extreme and hazardous events: to me, the innovation here is that, rather than using statistical methods that analyse time series of events that may arise from different processes, ClimaMeter considers changes in extreme weather arising from the same circulation patterns, which allows a more nuanced understanding of the spatial and multivariate changes associated with the weather type of interest. (This is particularly important for wind and precipitation, because unlike in univariate probabilistic attribution, the event definition doesn't require averaging over the spatial domain and so doesn't smooth out the local extremes)

**It was not our intention to imply that ClimaMeter is the only initiative communicating on extreme weather events in a changing climate. We revised this passage as suggested to acknowledge the fact that there are other complementary initiatives. We also better highlight, as noted by the Reviewer, the novel contribution that ClimaMeter brings to the field in terms of the focus on circulation patterns rather than extreme value statistics.**

410. Rather than referring to the four case studies, it would be more relevant to refer to the map in Figure 2.

**We modified it accordingly.**

411-412. I'm not sure what 'These underscore the significance of contextualising extreme events, as a tool to understand the broader context within which they occur' means here.

**We have shortened and restructured this part of the conclusions.**

490. This should be updated to 2001-2024 to be current at time of publication.

**We will change accordingly.**

Figures A1-A8. The values of ENSO, AMO and PDO associated with the event are missing from the plots.

**Thank you for spotting this, we have not added them as they are not used to compute the left-hand side dial**

================================================================================

**Anonymous Referee #2**

This manuscript presents a novel platform for assessing extreme events ("ClimaMeter"), which offers a useful database for past and future analyses. The authors present the protocol for ClimaMeter and explain its output and how the events are visualised. The presentation is clear and well-structured. In my opinion, the manuscript should be acceptable for publication after minor revisions.

General comments

In addition to reviewing the manuscript, I have read the report of Referee #1 and generally agree with their comments. In particular, I had the same confusion about the exact purpose of the paper. The discussion is very general, with explanations that could be reported on the ClimaMeter website, and does not have the specific details you would expect from a scientific paper. A more detailed description of (at least) one of the studies, as Referee #1 proposed, would certainly improve the scope of the paper.

**We thank the Reviewer for dedicating their time to assess our paper and for the positive feedback. As detailed in our replies to Reviewer #1, we plan to limit ourselves to two examples in the main paper, but including additional figures to provide a description and discussion that goes beyond what is reported on the ClimaMeter website.**

Specific comments

L53-55 – A bit more detail could be given on the event selection. Are there any plans on making this a more automatic process?

**At the present there is no plan for an automated process, and we admit that the event selection is also to some extent steered by the time availability of the core group members. However, we are**

**open for the future to use some automated event detection method (e.g., Latent Dirichlet Allocation as tested by some of the authors of this manuscript in a previous study) to select the events to be analysed. We added more details on this in the revised text.**

L72 – A climate reconstruction still uses a combination of observations and models; how is this independent of model bias?

**We agree that this is an imprecise wording. We rephrased it to "minimising the influence of model bias". Indeed, the whole idea of data assimilation is to limit the inherent biases of the model by "anchoring" it to observational data.**

L81 – It would be useful to report the actual grid size of MSWX here.

**We added this information in the revised version. The grid size is 0.1°x0.1°.**

L94 – It is unclear to me how these analogues are exactly defined. Do you ever consider similar events at other locations?

**Analogues are defined as those surface pressure maps displaying closest Euclidean distances with respect to the event itself within the analyzed domain. We consider an area of interest around the event and do not look for similar patterns at other locations. We specify this in the revised text. The latter choice is motivated by the fact that similar SLP patterns at different locations could have very different surface effects (in terms of temperature, wind and precipitation) even in a perfectly stationary climate, and this would thus bias our analysis.**

L140-145 – The way the gauge is presented makes it seem that the conclusion of natural variability vs climate change is very certain, but it is based on only three indices. You mention some of the drawbacks of this earlier in the text, but the final presentation (i.e. the left-hand gauge) can lead to the interpretation that an event is (for example) completely outside of natural variability, while there may be other factors playing a role. A more detailed description of the reasoning behind these choices will be helpful.

**In response to some of the comments from Reviewer #1 we discuss more systematically the methodological limitations of our approach in Sect. 2 and also added a paragraph on these in the conclusion section. We also now raise the point mentioned by the Reviewer (who is absolutely correct in their interpretation), giving land-surface changes as an example.**

Technical corrections

L17 – "Hurrícane" → "Hurricane"

L113 – "different" → "difference"

L136 – "or wind" → "and wind" (as you show them all)

L137 – ""past" and "present"" → ""present" and "past"" (as you show "present" − "past")

Fig. 1 – ""past" and "present"" → ""present" and "past"" (as you show "present" − "past")

L163 – "or wind" → "and wind" (as you show them all)
L209 – "analyzed" → "analysed"

**Thank you for spotting these. We corrected all these typos and technical errors in the revised text where they were still present.**

---

## Referee Report (RR1)

**Reviewer comments on 2023-2643-ATC1**

The authors have clearly taken my comments on board and made substantial revisions, and the focus of the paper – and the aim of the ClimaMeter methodology – is clearer as a result. I have a few additional comments below, mainly relating to the new text.

51. Replace 'at the basis of' with 'which forms the basis of'

59. 'a visual overview'

67. 'and Storm Poly'

86. Suggest 'Operationally we use data from MSWX' to make the distinction clearer

99. Typo – 'Finally, while'

134-5. It's not clear whether analogues within a window of events that last for more than one day are also excluded. Why are one-day events treated differently?

164. Punctuation: ClimaMeter (please also do a case-sensitive search to check for other instances of this)

191. I think the definition of Q is still a little unclear. Is Q the mean of the EDs or does each analogue have 15 Q values, one for each analogue? Suggest 'If … Q_obs (the mean ED from the actual event to the analogues) is below the 75$^{th}$ percentile…', and replacing Q with Q_obs where necessary.

210. Remove 'creating the corresponding difference maps' – I don't think maps of the bootstraps are produced, only the statndard deviations are used to identify significant differences.

213. Typo: 'is highlighted'

219. I think the addition of the 'statistics of events' here breaks up the flow of the description of the methodology, which would otherwise move smoothly from description of the analysis to the communication protocol. Suggest moving this to just before the conclusions, where it would fit better.

220. This explanation is not very clear. Suggest 'Figure 2a presents the median value of the gauge values over all 41 events studied by ClimaMeter to date; Figure 2b shows the proportion of events found to be influenced by each of the modes of natural climate variability considered'. Actually, I don't think this is a very clear way of displaying this information: both subfigures are bar plots, but one represents the median gauge value (and should therefore only be able to take values between 5 and 95%) while the other represents the proportion of events affected. Using a bar chart for the proportions, which can take any value between 0 and 100 and where the shaded area is meaningful, is fine. However, Figure 2a is potentially misleading: a better approach would be to use histograms showing the distribution of the gauge values. A more compact alternative would be to plot a bar from 5-95% for each column, then to shade the appropriate segment according to the number of occurrences; in which case, to avoid confusion over the meaning of the two plots, it might be better to show Figure 2a using horizontal bars, rather than vertical.

228-9. This isn't strictly accurate: Figure 2a doesn't show the full distribution, so it's not clear whether most studies have only 50% contribution from climate change, or whether half have 5%

contribution and the other half 95%. Furthermore, the ClimaMeter method doesn't actually evaluate the contribution from ACC directly, it only excludes the influence of other modes of variability: it would therefore be better to say something along the lines of 'the median percentage value suggests that differences between the analogues in the current and past periods can often be at least partially explained by modes of natural variability, rather than by climate change alone'.

231. It's worth mentioning that the finding that the climate change signal is most visible in heat extremes is in line with the IPCC's findings – see Figure SPM.3 (https://www.ipcc.ch/report/ar6/wg1/figures/summary-for-policymakers/figure-spm-3)

236. I don't see how this demonstrates the capabilities of ClimaMeter, although it's useful to have an overview of the results of past analyses – suggest removing the first part of the sentence and moving this paragraph to the end of the section.

375. 'the content is consistent with…'

413. Suggest 'than usual for the time of year' for clarity

420. 'weather situations' seems like an odd phrase here – maybe 'large-scale pressure events' or something similar?

423. Typo: 'the pressure over Britanny has become higher, while it has become lower over Italy'.

440. The conclusions here need to be updated.

485. Add reference to Figure A4

488. Given the uniqueness of the event, is it accurate to talk about 'storms similar to Poly'? What are the events that are identified as analogues?

492. Typo: Fig 8)

496-8. Typos: 'when comparing **the** gauge plots. Indeed, we **find** evidence that for the MSWX analogues the event **is** unique'

499. Could the selection of different analogues be due to choosing too small a domain size? Would it be better to consider a larger domain in order to find more consistent analogues? It would also be useful to know if differences in analogues are more common for certain types of weather events – this could be cited as potential future work if there's not enough information to judge at the moment.

504. Typo: '**and** the C3s…'

507. You should recap in the conclusions that, as mentioned in lines 99-100, there are limits to where analogues can be used effectively; this would also be a good place to acknowledge that the results may be sensitive to the choice of dataset and domain used.

520-1. I don't think this is necessarily a limitation of considering only the satellite era: this could also lead to overestimation of the effect of climate change since preindustrial times due to changes in the rate of local warming, but since the effect of climate change isn't actually estimated in ClimaMeter reports, that's not a major problem. I'd comment instead that the method risks underestimating the contribution of climate change by reducing the assumed influence of climate

change by 33% each time a significant difference is detected in a mode of natural variability, when actually both factors may contribute to the observed change in intensity.

532. Given the comment in lines 527-529 about ClimaMeter's role as an initial evaluation of the event, I think it's important not to oversell the potential use of the method in its rapid-attribution form in the next line. Suggest changing to 'Researchers can utilize ClimaMeter's methodology to…'

534. What kind of events not addressed in the literature?

538. Typo: 'to investigate the role…'

539-541. Again, the role of other attribution services should be acknowledged: confidence in any attribution result is increased if different methods produce similar or consistent conclusions. Suggest 'Policymakers can rely on ClimaMeter as an additional source of evidence as to how and to what extent specific extreme event categories in a given geographical area have changed over time thus enhancing the overall knowledge basis…' (since the changes are not, strictly speaking, attributed to climate change by this method).

541-544. This would follow on well from the end of the previous paragraph highlighting the benefit of ClimaMeter (and, I think, currently its main function) as a rapid communication tool, so I'd move this up to the start of this paragraph, and perhaps highlight again the speed with which reports can be produced.

561. (Perhaps more of an operational point) – it's important to distinguish between low confidence due to uniqueness of the event and low confidence due to inconsistency with previous results: for example, a very unique heatwave we would still be confident that climate change played a part, but under this framework the headline would be 'low confidence', which could be misleading.

Figures A1-A6. The observed values of ENSO, AMO and PDO are still missing.

697. Bibliography entry for Guardian article is wrongly formatted.

---

## Author Response (AR2)

Dear Dr. Ceppi,

We have completed the minor revisions requested by the reviewers for our manuscript titled ClimaMeter: Contextualising Extreme Weather in a Changing Climate. We have addressed all of Reviewer #1's comments as follows:

1. **Text Corrections:**
    - Corrected various typographical errors and improved phrasing as noted.
2. **Clarifications and Additions:**
    - Clarified data usage (Line 86).
    - Removed the sentence about excluding multi-day events (Lines 134-135).
    - Corrected all instances of "ClimaMeter" (Line 164).
    - Clarified the computation of metrics by introducing Q and Q_a (Line 191).
    - Updated figure presentations and descriptions, especially Figure 2a (Lines 210, 220).
    - Repositioned the "statistics of events" section for better flow (Line 219).
    - Revised discussions on natural variability and climate change contributions (Lines 228-229, 231).
    - Added limitations and potential future work considerations (Lines 499, 507).
    - Addressed the 33% reduction approach and potential biases (Lines 520-521).
    - Expanded on ClimaMeter's capabilities and role in attribution (Lines 534, 539-541).

We have ensured that all suggestions have been incorporated and that the manuscript is now more precise and clearer. Detailed answers are provided below.

Thank you for considering our revised manuscript.

Best regards,

Davide Faranda
On behalf of the ClimaMeter team

**Report #1**

The authors have clearly taken my comments on board and made substantial revisions, and the focus of the paper – and the aim of the ClimaMeter methodology – is clearer as a result. I have a few additional comments below, mainly relating to the new text.

51. Replace 'at the basis of' with 'which forms the basis of'

**corrected**

59. 'a visual overview'

**corrected**

67. 'and Storm Poly'

**corrected**

86. Suggest 'Operationally we use data from MSWX' to make the distinction clearer

**suggestion accepted**

99. Typo – 'Finally, while'

**corrected**

134-5. It's not clear whether analogues within a window of events that last for more than one day are also excluded. Why are one-day events treated differently?

**We have removed this sentence. This is a feature that we are still implementing and that was not applied to the case studies shown in this paper. The exclusion protocol will be applied to all events, including those lasting longer than one day, in further studies and versions of ClimaMeter.**

164. Punctuation: ClimaMeter (please also do a case-sensitive search to check for other instances of this)

**We have corrected all the instances ClimaMeter with the right spelling**

191. I think the definition of Q is still a little unclear. Is Q the mean of the EDs or does each analogue have 15 Q values, one for each analogue? Suggest 'If … $Q\_obs$ (the mean ED from the actual event to the analogues) is below the $75^{th}$ percentile…', and replacing Q with $Q\_obs$ where necessary.

**We thank the reviewer for this suggestion. Indeed, the previous formulation was confusing. We have now introduced two quantities. We leave Q as the analogue quality and name Q_a the full distribution of Euclidean distances of the best analogues of the event. Introducing Q_a makes the computation of the metric more understandable.**

210. Remove 'creating the corresponding difference maps' – I don't think maps of the bootstraps are produced, only the standard deviations are used to identify significant differences.

**corrected**

213. Typo: 'is highlighted'

**corrected**

219. I think the addition of the 'statistics of events' here breaks up the flow of the description of the methodology, which would otherwise move smoothly from description of the analysis to the communication protocol. Suggest moving this to just before the conclusions, where it would fit better.

**As suggested, we have moved this section before the conclusions.**

220. This explanation is not very clear. Suggest 'Figure 2a presents the median value of the gauge values over all 41 events studied by ClimaMeter to date; Figure 2b shows the proportion of events found to be influenced by each of the modes of natural climate variability considered'.
Actually, I don't think this is a very clear way of displaying this information: both subfigures are bar plots, but one represents the median gauge value (and should therefore only be able to take values between 5 and 95%) while the other represents the proportion of events affected. Using a bar chart for the proportions, which can take any value between 0 and 100 and where the shaded area is meaningful, is fine. However, Figure 2a is potentially misleading: a better approach would be to use histograms showing the distribution of the gauge values. A more compact alternative would be to plot a bar from 5-95% for each column, then to shade the appropriate segment according to the number of occurrences; in which case, to avoid confusion over the meaning of the two plots, it might be better to show Figure 2a using horizontal bars, rather than vertical.

**We agree with the reviewer's comment and we now present (ex Figure 2a and now Figure 9) as an error-bar plot. The new figure shows error-bars reporting the median values (circles) and the standard deviation (whiskers) of ClimaMeter's climate change and uniqueness gauges. We have consequently updated the description of the figure in the text.**

[Figure]

228-9. This isn't strictly accurate: Figure 2a doesn't show the full distribution, so it's not clear whether most studies have only 50% contribution from climate change, or whether half have 5% contribution and the other half 95%. Furthermore, the ClimaMeter method doesn't actually evaluate the contribution from ACC directly, it only excludes the influence of other modes of variability: it would therefore be better to say something along the lines of 'the median percentage value suggests that differences between the analogues in the current and past periods can often be at least partially explained by modes of natural variability, rather than by climate change alone'.

**Thanks for this important comment. We have changed this sentence and the discussion of Fig. 2 along the lines suggested by the reviewer.**

231. It's worth mentioning that the finding that the climate change signal is most visible in heat extremes is in line with the IPCC's findings – see Figure SPM.3 (https://www.ipcc.ch/report/ar6/wg1/figures/summary-for-policymakers/figure-spm-3)

**Thank you for the suggestion, this has been updated as suggested.**

236. I don't see how this demonstrates the capabilities of ClimaMeter, although it's useful to have an overview of the results of past analyses – suggest removing the first part of the sentence and moving this paragraph to the end of the section.

**We have modified the sentence accordingly**

375. 'the content is consistent with…'

**corrected**

413. Suggest 'than usual for the time of year' for clarity

**corrected**

420. 'weather situations' seems like an odd phrase here – maybe 'large-scale pressure events' or something similar?

**corrected**

423. Typo: 'the pressure over Britanny has become higher, while it has become lower over Italy'.

**corrected**

440. The conclusions here need to be updated.

**updated**

485. Add reference to Figure A4

**added**

488. Given the uniqueness of the event, is it accurate to talk about 'storms similar to Poly'? What are the events that are identified as analogues?

**The identified patterns consist of low pressure systems with weaker anomalies or displaced spatially; this has been added to the text.**

492. Typo: Fig 8)

**corrected**

496-8. Typos: 'when comparing the gauge plots. Indeed, we find evidence that for the MSWX analogues the event is unique'

**corrected**

499. Could the selection of different analogues be due to choosing too small a domain size? Would it be better to consider a larger domain in order to find more consistent analogues? It

would also be useful to know if differences in analogues are more common for certain types of weather events – this could be cited as potential future work if there's not enough information to judge at the moment.

**These limitations are now clearly stated, see our answer to the reviewer comment for LL 539-541 indicated by (\*)**

504. Typo: 'and the C3s…'

**corrected**

507. You should recap in the conclusions that, as mentioned in lines 99-100, there are limits to where analogues can be used effectively; this would also be a good place to acknowledge that the results may be sensitive to the choice of dataset and domain used.

**These limitations are now clearly stated, see our answer to the reviewer comment for LL 539-541 indicated by (\*)**

520-1. I don't think this is necessarily a limitation of considering only the satellite era: this could also lead to overestimation of the effect of climate change since preindustrial times due to changes in the rate of local warming, but since the effect of climate change isn't actually estimated in ClimaMeter reports, that's not a major problem. I'd comment instead that the method risks underestimating the contribution of climate change by reducing the assumed influence of climate change by 33% each time a significant difference is detected in a mode of natural variability, when actually both factors may contribute to the observed change in intensity.

**Thank you for the suggestion, we have modified the text accordingly and now concerning the analysis period chosen, we highlight the issue with the brevity of the time series rather than specifying in which direction potential biases that this introduces would go. We have further added a comment on the approach of reducing the assumed influence of climate change by 33% each time a significant difference is detected in a mode of natural variability, noting that this indeed risks underestimating the true contribution of climate change.**

532. Given the comment in lines 527-529 about ClimaMeter's role as an initial evaluation of the event, I think it's important not to oversell the potential use of the method in its rapid-attribution form in the next line. Suggest changing to 'Researchers can utilize ClimaMeter's methodology to…'

**corrected**

534. What kind of events not addressed in the literature?

**We modified the sentence as follows:**
**"Researchers can utilize ClimaMeter's methodology to delve into the relationship between climate change and specific extreme events, even those not typically addressed in statistical attribution studies, such as medicanes, explosive extratropical cyclones, tropical cyclones, Acqua Alta events in Venice and others."**

538. Typo: 'to investigate the role…'

**corrected**

539-541. Again, the role of other attribution services should be acknowledged: confidence in any attribution result is increased if different methods produce similar or consistent conclusions. Suggest 'Policymakers can rely on ClimaMeter as an additional source of evidence as to how and to what extent specific extreme event categories in a given geographical area have changed over time thus enhancing the overall knowledge basis…' (since the changes are not, strictly speaking, attributed to climate change by this method).

**(*) Thank you for the comment, we agree with these considerations. We have rephrased the passage where we discuss the use of ClimaMeter by policy makers. We have additionally included the following paragraph in the conclusions:**
**"Related to this, we also reiterate that the spatial domain used for the analogues is chosen based on expert judgment from members of the ClimaMeter project, and therefore it carries an arbitrary component. The results of a ClimaMeter analysis are likely to provide different results if different domains are chosen, especially if larger or smaller-scale features become dominant due to a much larger or smaller domain. Furthermore, we stress that there are limits to the analogues approach. For example, conclusions about the impact of climate change are more robust in case studies where the analogue quality is high, and the long-term climate trends match the changes in the extreme event itself. Finally, ClimaMeter is designed to work for extreme events whose dynamics can be well represented through circulation analogues. Small-scale events where local processes are important – e.g. an isolated tornado or a hail storm – are currently outside the scope of ClimaMeter."**

541-544. This would follow on well from the end of the previous paragraph highlighting the benefit of ClimaMeter (and, I think, currently its main function) as a rapid communication tool, so I'd move this up to the start of this paragraph, and perhaps highlight again the speed with which reports can be produced.
**As suggested, we have repeated the advantage of ClimaMeter in terms of rapidity for producing quick reports on extreme weather events in the conclusions**

561. (Perhaps more of an operational point) – it's important to distinguish between low confidence due to uniqueness of the event and low confidence due to inconsistency with previous results: for example, a very unique heatwave we would still be confident that

climate change played a part, but under this framework the headline would be 'low confidence', which could be misleading.

**We have this point well in mind for our "operational" approaches. Yet we believe that a single confidence statement should be provided in the report for clarity of communication with the media and general public and that it is better to err on the side of caution rather than potentially overstating the confidence we have in our results.**

Figures A1-A6. The observed values of ENSO, AMO and PDO are still missing .

**Thanks again for this suggestion. However, as mentioned in the previous answer for revision 1, we will not add this information to the supplementary ClimaMeter figure or to this paper. In most cases, ClimaMeter covers weather extremes immediately after the event, so that monthly teleconnection indices are not usually available at the time of the analysis. Furthermore, the observed values of ENSO, AMO, and PDO are not used to compute any statistics for ClimaMeter or to write the text report - we only use the statistical difference (or lack thereof) of the two ENSO/AMO/PDO distributions.**

697. Bibliography entry for Guardian article is wrongly formatted.

**This reference no longer appears in the text.**

**Report #2**

Accept as it is